# The main oxidative inactivation pathway of the plant hormone auxin

Ken-ichiro Hayashi [1✉], Kazushi Arai [1], Yuki Aoi [2], Yuka Tanaka [1], Hayao Hira [3], Ruipan Guo [4], Yun Hu [4], Chennan Ge [4], Yunde Zhao [4], Hiroyuki Kasahara [3,5] & Kosuke Fukui [1]

Inactivation of the phytohormone auxin plays important roles in plant development, and several enzymes have been implicated in auxin inactivation. In this study, we show that the predominant natural auxin, indole-3-acetic acid (IAA), is mainly inactivated via the GH3-ILR1-DAO pathway. IAA is first converted to IAA-amino acid conjugates by GH3 IAA-amidosynthetases. The IAA-amino acid conjugates IAA-aspartate (IAA-Asp) and IAA-glutamate (IAA-Glu) are storage forms of IAA and can be converted back to IAA by ILR1/ILL amidohydrolases. We further show that DAO1 dioxygenase irreversibly oxidizes IAA-Asp and IAA-Glu into 2-oxindole-3-acetic acid-aspartate (oxIAA-Asp) and oxIAA-Glu, which are subsequently hydrolyzed by ILR1 to release inactive oxIAA. This work established a complete pathway for the oxidative inactivation of auxin and defines the roles played by auxin homeostasis in plant development.

[1] Department of Biochemistry, Okayama University of Science, Okayama 700-0005, Japan. [2] Department of Biological Production Science, United Graduate School of Agricultural Science, Tokyo University of Agriculture and Technology, Fuchu, Tokyo 183-8509, Japan. [3] Graduate School of Agriculture, Tokyo University of Agriculture and Technology, Fuchu, Tokyo 183-8509, Japan. [4] Section of Cell and Developmental Biology, University of California San Diego, Gilman Dr. La Jolla, San Diego, CA 92093-0116, USA. [5] RIKEN Center for Sustainable Resource Science, Yokohama, Kanagawa 230-0045, Japan. ✉email: hayashi@dbc.ous.ac.jp

The phytohormone auxin plays a pivotal role in almost every aspect of plant development. Auxin biosynthesis, polar transport, and inactivation pathways coordinately modulate cellular auxin homeostasis in plants. Indole-3-acetic acid (IAA) is the predominant natural auxin and is synthesized from tryptophan via two consecutive enzymatic reactions in the indole-3-pyruvate pathway, which is widely conserved in land plants[1]. In *Arabidopsis*, four types of IAA-inactivating enzymes have been reported (Fig. 1 and Supplementary Fig. 1a)[2–4]. IAA CARBOXYL METHYLTRANSFERASE1 (IAMT1) converts IAA to IAA methyl ester[5,6]. The UDP-glucosyltransferase UGT84B1 catalyzes the formation of IAA-β-D-glucoside in vivo and in vitro[7,8]. Both IAA methyl ester and IAA-glucoside are considered to be biologically inactive. The *iamt1* and *ugt84b1* null mutants exhibited no obvious auxin overaccumulation phenotypes in *Arabidopsis* plants[6,8]. Oxidation of IAA has long been believed to be a major IAA inactivation pathway because 2-oxindole-3-acetic acid (oxIAA) is the most abundant IAA metabolite in *Arabidopsis* plants[3,9]. *DIOXYGENASE FOR AUXIN OXIDATION 1* (*DAO1*) is a member of the 2-oxoglutarate and Fe(II)-dependent oxygenase superfamily. Both *Arabidopsis* AtDAO1 and rice OsDAO convert IAA to oxIAA in vitro[10–12]. Although the *Arabidopsis* loss-of-function *dao1-1* mutant did not produce oxIAA, it accumulated IAA-*L*-aspartate (IAA-Asp) and IAA-*L*-glutamate (IAA-Glu) conjugates, which was hypothesized to be a compensatory route for the reduced IAA degradation in *dao1* mutants[10]. The *dao1-1* mutants exhibited auxin-accumulation phenotypes, but *AtDAO1*-overexpressing plants did not confer obvious auxin-deficient phenotypes[10,11].

*GH3* genes encode the acyl amidosynthetases that catalyze the conjugation reactions of salicylic acid, jasmonic acid, and IAA with amino acids. *GH3* genes are classified into three groups (I to III), and eight *GH3* genes in group II redundantly inactivate auxin in *Arabidopsis* plants[13]. Six *GH3* genes (*GH3.1–GH3.6* in group II) are auxin-inducible genes that mainly produce IAA-Asp, suggesting that they play regulatory roles in attenuating active auxin concentrations[13]. The expression of *GH3.9* and *GH3.17* in group II is not dependent on auxin, and the encoded enzymes catalyze the formation of IAA-Glu, suggesting that they primarily function in basal IAA inactivation[13,14]. *GH3.17* is involved in

shade-induced hypocotyl elongation by regulating the IAA level[15]. The *GH3.6*-, *UGT84B1*-, and *IAMT1*-overexpressing plants showed severe auxin-deficient phenotypes[6,8,16]. In contrast, loss-of-function mutants for the individual IAA-inactivating enzymes exhibited only slightly impaired or normal phenotypes in auxin-regulated development processes[6,8,13], consistent with the redundant roles by these enzymes and pathways in maintaining IAA homeostasis.

*IAA-Leu-Resistant1* (*ILR1*) was initially identified as an IAA-Leu amidohydrolase from a genetic screen for mutants resistant to IAA-Leu[17]. The *Arabidopsis ILR1-like* (*ILL*) family consists of *ILR1*, *ILL1*, *ILL2*, *ILL3*, *IAR3/ILL4*, *ILL6*, and the pseudogene *ILL5*[18–20]. ILR1/ILL enzymes convert various IAA-amino acid conjugates to IAA in vitro. However, whether the ILR1/ILL enzymes are involved in metabolizing IAA-Asp and IAA-Glu in plants has not been experimentally tested because both conjugates did not trigger auxin-phenotypes when added to plant growth media[20].

In this study, we demonstrate that IAA is mainly inactivated by GH3 enzymes and that DAO functions as an oxidase of IAA-amino acid conjugates to produce oxIAA-amino acid conjugates downstream of GH3. Using the membrane-permeable ester of IAA-amino acid conjugates, we demonstrate that IAA-Asp and IAA-Glu are reversible storage forms of IAA, contrary to the previously proposed function of the conjugates. ILR1/ILL enzymes revert IAA-Asp and IAA-Glu back to free IAA in planta. Unlike the widely accepted pathway[3,4], oxIAA is produced from oxIAA-amino acids by ILR1 but not by direct oxidation of IAA. Thus, we determine that IAA inactivation is coordinately regulated by GH3-ILR1-DAO-mediated IAA recycling and degradation to maintain auxin homeostasis. Our findings provide insights into the regulatory framework governing auxin homeostasis.

## Results

**GH3 pathway plays a central role in IAA inactivation**. To determine the contributions of each IAA-inactivating enzyme to IAA homeostasis, we generated multiple null mutants by CRISPR-Cas9 gene editing technology. *Arabidopsis dao1 dao2* double mutants and *iamt1 ugt84b1 dao1 dao2* quadruple mutants showed primary root growth similar to that of wild-type (WT)

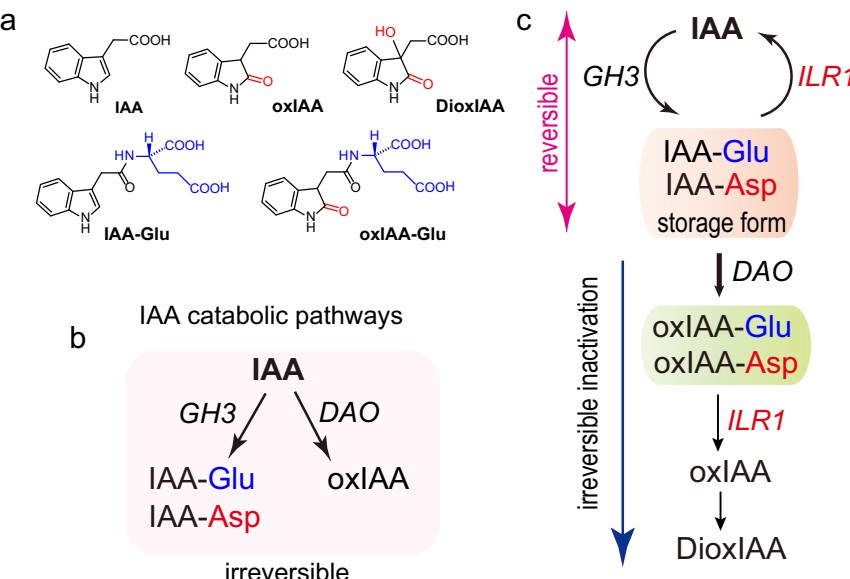

**Fig. 1 Structures of IAA metabolites and IAA inactivation pathways. a** Structures of IAA metabolites. **b** The previous model of IAA catabolic pathways. GH3 and DAO irreversibly inactivate IAA through different pathways. **c** The new model proposed here: the reversible IAA inactivation pathway composed of GH3, ILR1, and DAO1 in *Arabidopsis* plants.

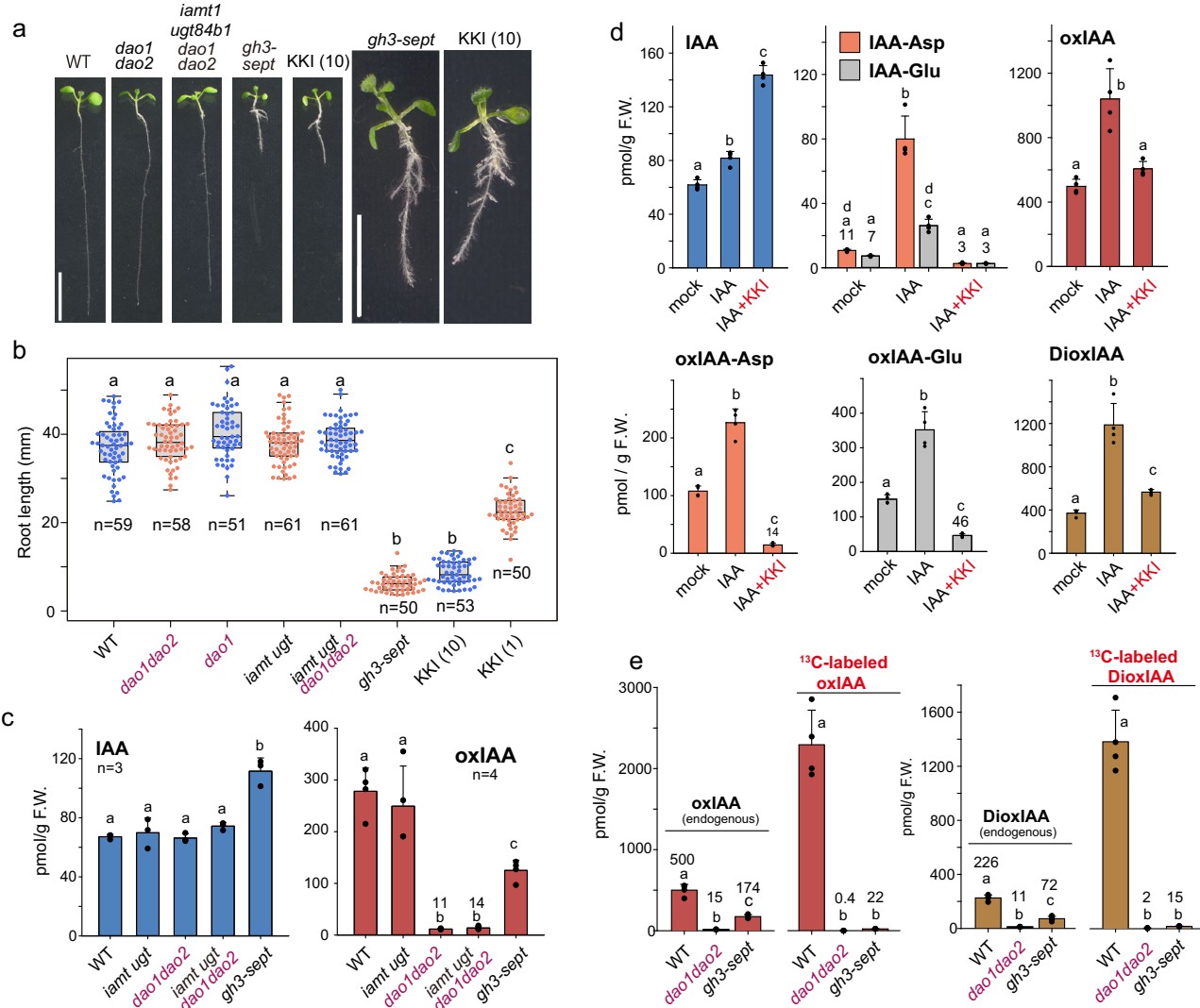

**Fig. 2 GH3 plays a central role in IAA inactivation. a**, **b** Phenotypes of the IAA-inactivating enzyme mutants *dao1 dao2*, *iamt1 ugt84b1 dao1 dao2*, and *gh3-1 2 3 4 5 6 17* (*gh3-sept*) as well as WT (Col) plants grown with the GH3 inhibitor kakeimide (KKI) on GM plates for 8 days. Scale bar, 10 mm. The length of 8-d-old root was measured. Box plots show the median as lines in the box, the 1st to 3rd quartiles as bounds of box, whiskers with end caps extending 1.5-fold interquartile range beyond the box, and the end caps represent the minimum and maximum. The different letters represent statistical significance at $P < 0.001$ (Tukey's HSD test, $n = 50$–61). The values in parentheses represent μM. **c** The levels of endogenous IAA and oxIAA in IAA-inactivating enzyme mutants (8-d-old). **d** The levels of IAA metabolites after cotreatment with 0.2 μM IAA and 20 μM GH3 inhibitor (KKI). WT seedlings (7-d-old) were incubated with IAA and KKI for 20 h. **e** [$^{13}C_6$]IAA feeding experiment. [$^{12}C$]-endogenous and [$^{13}C_6$]-labeled oxIAA and dioxIAA were measured using [$^2H_2$]oxIAA and [$^2H_2$]dioxIAA as internal standards after incubation of 7-d-old WT, *dao1dao2*, and *gh3-sept* seedlings with 0.2 μM [$^{13}C_6$]IAA for 20 h. **c**–**e** The values shown are the means ± SD (**d**, e, $n = 4$). The different letters represent statistical significance at $P < 0.01$ (Tukey's HSD test).

plants (Fig. 2a, b and Supplementary Fig. 1b). On the other hand, *gh3.1 2 3 4 5 6 17* septuple (*gh3-sept*) mutants exhibited very short roots and more adventitious roots, which are characteristic phenotypes of auxin overaccumulation[21]. We generated a GH3 inhibitor designated kakeimide (KKI) that competitively inhibited IAA conjugation (Supplementary methods)[22]. WT seedlings treated with KKI phenocopied *gh3-sept* mutants which were insensitive to KKI treatments (Fig. 2a and Supplementary Fig. 2a). Additionally, cotreatment with IAA and KKI synergistically enhanced auxin-inducible *DR5::GUS* reporter gene expression (Supplementary Fig. 2b) and KKI enhanced the high-auxin phenotypes in the *35S::YUC1* lines which overproduce auxin (Supplementary Fig. 2c), suggesting that KKI elevated endogenous IAA levels by inhibiting group II GH3 activities. The *dao1 dao2* mutants showed a slightly increased sensitivity to IAA in assays of primary root growth compared with that of WT

(Supplementary Fig. 2d). On the other hand, the *iamt1 ugt84b1 dao1 dao2* mutants showed the same sensitivity to IAA and KKI as the *dao1 dao2* mutants, suggesting that *IAMT1* and *UGT84B1* do not play major roles in IAA homeostasis in vivo (Supplementary Fig. 2d, e).

Consistent with the high-auxin phenotypes, endogenous IAA levels were elevated, and IAA-Asp and IAA-Glu were undetected, in the *gh3-sept* mutants (Fig. 2c and Supplementary Fig. 3a). KKI caused a significant accumulation of IAA by inhibiting the conversion of IAA to IAA-Asp and IAA-Glu in WT (Fig. 2d and Supplementary Fig. 3d). Furthermore, KKI also led to a high accumulation of IAA and repressed the conversion of IAA to IAA-Asp and IAA-Glu in *YUC2*-overexpressing auxin-overproduction plants (Supplementary Fig. 3d). Endogenous IAA levels did not differ significantly between *iamt1 ugt84b1 dao1 dao2* mutants and WT plants (Fig. 2c). These results

indicate that the GH3 pathway plays a central role in IAA inactivation.

**DAO functions as an IAA-amino acid oxidase.** The previous model suggests that *AtDAO1* is involved in basal IAA inactivation[23]. When IAA levels are elevated, GH3s convert IAA to IAA-Asp and IAA-Glu to maintain IAA homeostasis (Fig. 1b). *GH3* expression is induced in the *dao1-1* loss-of-function mutant in response to reduced IAA degradation, resulting in extreme accumulation of the two conjugates in *dao1-1* (Supplementary Fig. 3a, d)[10]. Thus, we expected that oxIAA would accumulate to high levels in *gh3-sept* mutants. Surprisingly, oxIAA was significantly decreased in *gh3-sept* mutants (Fig. 2c), and KKI repressed oxIAA production from IAA in *YUC2*-overexpressing and WT plants (Supplementary Fig. 3d and Fig. 2d). It is known that dioxIAA (3-hydroxy-oxIAA) is an oxidized product of oxIAA[24]. KKI also inhibited dioxIAA production from IAA in WT (Fig. 2d). DioxIAA was significantly decreased in both the *dao1 dao2* and *gh3-sept* mutants (Supplementary Fig. 3c). Feeding experiments using [$^{13}C_6$]-labeled IAA confirmed that exogenous [$^{13}C_6$]IAA was not converted to [$^{13}C_6$]oxIAA and [$^{13}C_6$]dioxIAA in *dao1 dao2* and *gh3-sept* mutants, but was efficiently converted in WT plants (Fig. 2e). These results indicate that the production of oxIAA depends on the presence of GH3 enzymes and that AtDAO1 likely functions downstream of GH3 in the same pathway (Fig. 1c).

To assess the substrate specificity of DAO enzymes, we studied the enzymatic activity of recombinant *Arabidopsis* AtDAO1 and rice OsDAO. AtDAO1 efficiently oxidized IAA-Asp, IAA-Glu, IAA-alanine (IAA-Ala), and IAA-leucine (IAA-Leu) to the corresponding oxIAA-amino acid conjugates. The Km values of AtDAO1 for IAA-Glu and IAA-Asp were 9.4 and 13.5 μM, respectively (Fig. 3a and Supplementary Fig. 4a–d). AtDAO1 did not recognize IAA-glucoside or IAA-Asp dimethyl ester (Supplementary Fig. 4e). Under our assay conditions, the AtDAO1 oxidation activity for IAA-Glu was 3.4 μmol/min/mg, whereas the oxidation activity for IAA was only 5.5 pmol/min/mg, indicating that IAA-Glu oxidation is approximately $6 \times 10^5$ times faster than IAA oxidation (Supplementary Fig. 4f). The crystal structures of AtDAO1 and OsDAO were recently reported[25,26]. Our molecular docking calculation for AtDAO1 and its substrates (IAA and IAA-amino acids) indicates that AtDAO1 prefers IAA-amino acids to IAA. IAA-Asp and IAA-Glu fit the active site of AtDAO1 better than IAA (Supplementary Fig. 5). OsDAO[12] also oxidized IAA-Asp and IAA-Glu to yield oxIAA-Asp and oxIAA-Glu, respectively, in vitro, but this enzyme did not produce oxIAA under the same conditions (Supplementary Fig. 6). Interestingly, OsDAO did not oxidize IAA-Ala or IAA-Leu under our assay conditions. These results demonstrate that AtDAO1 and OsDAO primarily function as IAA-amino acid oxidases, suggesting that the auxin-related phenotypes of *dao1-1* are caused by IAA release from IAA-Asp and IAA-Glu but not by the loss of IAA oxidation (Fig. 1c).

**IAA-Asp and IAA-Glu function as IAA storage forms in vivo.** IAA-Asp and IAA-Glu have long been suggested to be irreversible metabolites in planta because these two conjugates lack auxin activity when applied to *Arabidopsis* plants[4,20]. However, both IAA-Asp and IAA-Glu are hydrophilic IAA conjugates with dicarboxylic acids and thus might not be incorporated into cells due to poor membrane permeability. To test this hypothesis, we synthesized membrane-permeable prohormone IAA-Asp diesters as the prodrug form of IAA-Asp. We found that diesters acted as prodrugs of IAA-Asp and IAA-Glu to release free IAA, thereby exhibiting auxin activities in planta (Supplementary Fig. 7a, b). On the other hand, a monoester of IAA-Asp showed weak auxin

activity, perhaps due to a lower membrane permeability. To confirm that the IAA-amino acid dimethyl esters are metabolically converted into IAA, we analyzed the metabolites from [$^2H_2$]-labeled IAA-Asp-dimethyl ester ([$D_2$]IAA-Asp-DM: $^2H_2$-methylene group in IAA moiety) in *Arabidopsis* plants. The unmodified [$D_2$]IAA-Asp was detected in WT plants after treatment with [$D_2$]IAA-Asp-DM (Fig. 3b). [$D_2$]IAA appeared in WT plants, but $D_2$-IAA was not detectable in *ilr1 ill2 iar3 ill3* mutants after incubation with $D_2$-IAA-Asp-DM (Fig. 3b), indicating that ILR1/ILL hydrolases converted IAA-amino acid to free IAA in planta. These results illustrated that plasma membrane-permeable forms of IAA conjugates, IAA-Asp-DM and IAA-Glu dimethyl ester (IAA-Glu-DM), were metabolically converted to the unmodified IAA-Asp and IAA-Glu, and then ILR1/ILL enzymes hydrolyze the two conjugates to release free IAA. Consequently, the released IAA elicited typical auxin responses, such as root growth inhibition and lateral root promotion (Fig. 3c and Supplementary Fig. 7–9). IAA-Asp-DM and IAA-Glu-DM induced the expression of auxin-responsive *DR5* reporter gene (Supplementary Fig. 7c). Auxin-resistant signaling mutants were insensitive to IAA-amino acid diesters (Supplementary Fig. 7d). Furthermore, IAA-Asp-DM and IAA-Glu-DM inhibited root growth in the monocot plants *Oryza sativa* (rice) and *Brachypodium distachyon*. (Supplementary Fig. 8). These results demonstrate that IAA-Asp and IAA-Glu function as IAA storage forms in planta and are the substrates of the ILR1/ILL hydrolases. Other synthetic methyl esters of IAA-Leu, IAA-Ala, IAA-Phe, and IAA-Met also function as prodrugs of IAA-amino acid conjugates that were converted to free IAA in *Arabidopsis* plants (Supplementary Fig. 10a). On the other hands, endogenous IAA-Ala and IAA-Leu were not detected in the *dao1-1* mutant and WT plants in our measurements (Supplementary Fig. 10b), suggesting that these conjugates are minor metabolites in the IAA catabolic pathway.

**DAO oxidizes IAA-Asp and IAA-Glu in planta.** The *AtDAO1*-overexpressing lines (*35S::AtDAO1*) were highly resistant to IAA-amino acids and their esters, including IAA-Asp-DM and IAA-Glu-DM (Fig. 3c and Supplementary Fig. 9a, c, d and 10a, c, d), but showed the same response to IAA as WT (Supplementary Fig. 9e), suggesting that DAO1 rapidly converts IAA-Asp and IAA-Glu to inactive oxIAA-amino acids but does not efficiently oxidize IAA in planta. In contrast, the *dao1-1* mutant was hypersensitive to IAA-amino acid methyl ester (Fig. 3c and, Supplementary Fig. 9a, c, d and 10a, c, d). Additionally, unmodified IAA-Glu and IAA-Asp showed auxin activity in root growth and *DR5* transgene induction in *dao1* mutant (Fig. 3d and Supplementary Fig. 7c). Furthermore, the estradiol-inducible *AtDAO1* transgene (*pMDC7::AtDAO1*) complemented the high sensitivity to IAA-Asp-DM exhibited by the *dao1-1* mutant (Supplementary Fig. 9b). Overexpression of native and GFP-fused *O. sativa* OsDAO and *B. distachyon* BdDAO decreased the sensitivity to IAA-amino acid diesters in both WT (Fig. 4a, b) and *dao1-1* mutants (Supplementary Fig. 11a). The *dao1-1* mutants showed impaired fertility and shorter siliques with an irregular pattern in the primary inflorescence[10,11]. This impaired phenotype in *dao1-1* was complemented by *35S::GFP-AtDAO1* and *35S::GFP-OsDAO* (Fig. 4c). On the other hand, overexpression of GFP-AtDAO2 and the loss-of-function *dao2-1* mutation did not affect the sensitivity to IAA-Glu-DM (Fig. 4a, b and Supplementary Fig. 11b).

Consistent with phenotypic and biochemical studies, oxIAA-Asp and oxIAA-Glu were significantly decreased in the *dao1* mutants (Fig. 3e and Supplementary Fig. 3b). The *35S::GFP-AtDAO1*-overexpressing lines in *dao1-1* mutants increased

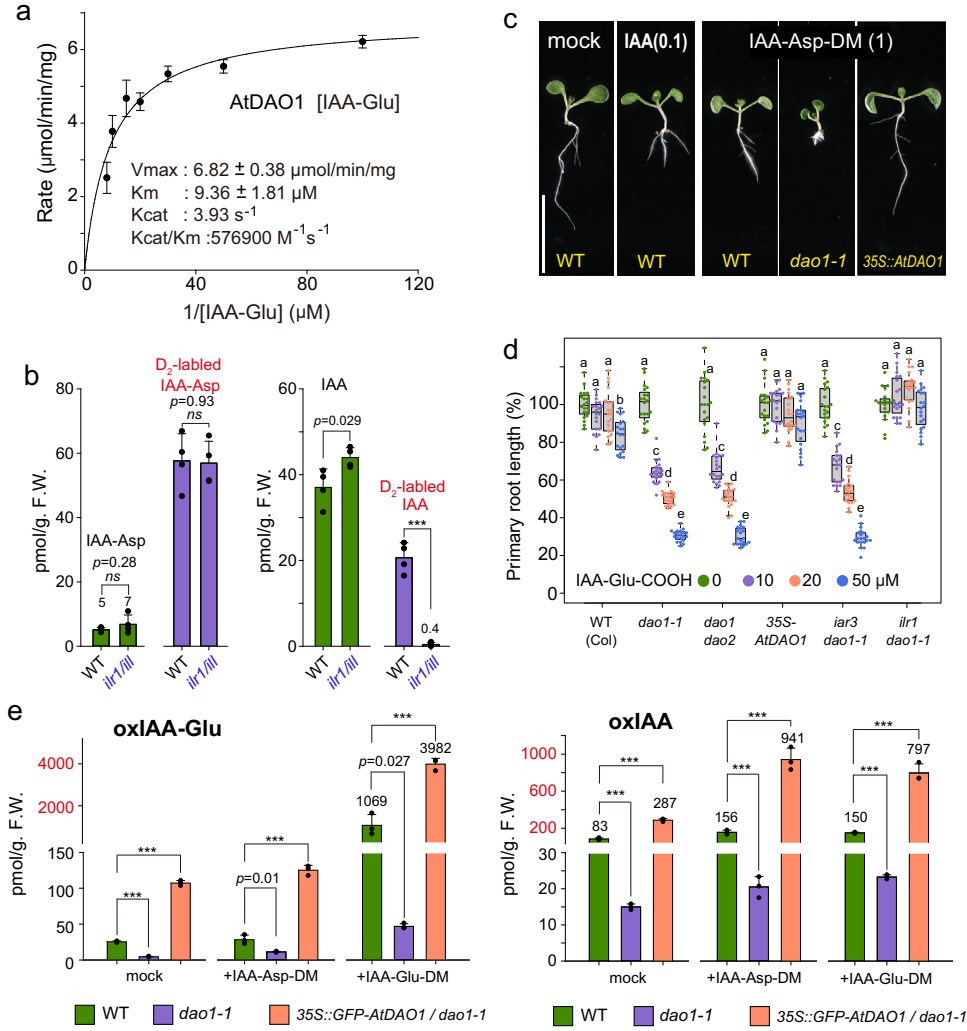

**Fig. 3 DAO1 irreversibly oxidized IAA-amino acid conjugates in a plant. a** Michaelis–Menten plot and kinetics parameters of the AtDAO1 enzyme for IAA-Glu ($n = 3$). Error bars, SE. **b** in vivo conversion of IAA-Asp-DM to IAA. Endogenous and [$D_2$]IAA-Asp and [$D_2$]IAA were measured after incubation of 8-d-old WT and *ilr1 ill2 iar3 ill3* (*ilr1/ill*) quadruple mutants with 1 μM [$D_2$]IAA-Asp-DM for 6 h. The values shown are the means ± SD (***$P < 0.001$, *ns*: not significant, two-tailed Student's *t* test, $n = 4$). **c** Phenotypes of *dao1-1*- and AtDAO1-overexpressing (*35 S::AtDAO1*) plants grown for 6 days with 1 μM IAA-Asp-DM. Scale bar, 10 mm. **d** Effects of IAA-Glu on root growth in *dao1-1*, *dao1 dao2*, *35 S::AtDAO1*, *iar3 dao1-1*, and *ilr1 dao1-1* mutants. Root length (6-d-old) is shown as the percentage of that in mock-treated plants (100%). The different letters represent statistical significance at $P < 0.001$ (Tukey's HSD test, $n = 20$). **e** The levels of IAA metabolites after treatment with IAA-amino acid diesters. WT, *dao1-1*, and *GFP-AtDAO1*-overexpressing *dao1-1* mutants (7-d-old) were incubated for 20 h with the compounds. The values shown are the means ± SD (***$P < 0.001$, two-tailed Student's *t* test, $n = 3$).

production of oxIAA, dioxIAA and their conjugates. The oxIAA metabolites accumulated after IAA-amino acid diester treatment in the WT and *AtDAO1* overexpression lines but not in the *dao1-1* mutant (Fig. 3e and Supplementary Fig. 12). As observed for the *dao1* mutants, the *gh3-sept* mutants also did not produce oxIAA-Asp and oxIAA-Glu (Supplementary Fig. 3b), and the GH3 inhibitor KKI blocked the synthesis of oxIAA-Asp and oxIAA-Glu in WT plants (Fig. 2d). Similarly, oxIAA conjugates and dioxIAA were highly accumulated in both *O. sativa* and *B. distachyon* seedlings after IAA and IAA-amino acid diester treatment (Supplementary Fig. 13). Notably, oxIAA was a minor metabolite in these monocot plants. oxIAA might be readily converted to dioxIAA in *O. sativa* and *B. distachyon*. These findings further indicate that AtDAO1 and its DAO1 orthologs primarily function as IAA-amino acid oxidases in vivo.

**ILR1/ILL regulate IAA homeostasis in planta.** To investigate the physiological role of the *ILR/ILL* genes (Supplementary Fig. 1b), the sensitivity of *ilr1/ill* mutants to IAA-amino acid diesters was

examined. Among the single mutants, only *ilr1* showed resistance to IAA-Asp-DM and IAA-Glu-DM in primary root growth and lateral root promotion (Fig. 5a and Supplementary Fig. 14, 15d). The *ilr1 iar3* double mutants were more resistant to these conjugate esters, and the *ilr1 ill2 iar3 ill3* quadruple mutants were insensitive to these conjugate esters at the tested concentrations (Supplementary Fig. 15). We also analyzed the expression of *ILR1-, ILL2-* and *IAR3-GFP/GUS* fusion proteins under their native promoters (Fig. 5b and Supplementary Fig. 16). These fusion proteins complemented the insensitivity of the *ilr1* and *iar3* mutants to IAA-amino acid diesters (Supplementary Fig. 16a, b). The ILR1-GFP/GUS and IAR3-GFP/GUS fusion proteins were expressed at the hypocotyl, lateral root tip, and meristem zone of the root and shoot, where IAA is constitutively synthesized and metabolized. The ILL2-GUS protein was also detected at the shoot meristem (Supplementary Fig. 16j). These expression patterns are consistent with the notion that the *ILR1/ILL* genes participate in IAA release from IAA-Asp and IAA-Glu. The *ilr1 ill2 iar3* triple mutants in the Col background showed

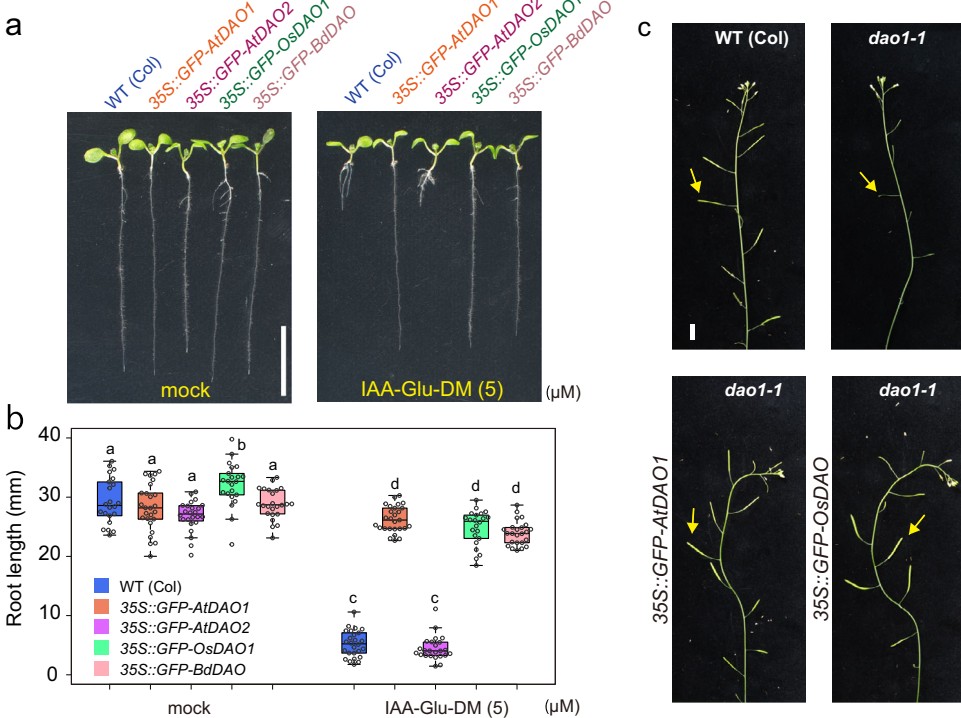

**Fig. 4 Monocot *O. sativa* and *B. distachyon* DAO enzymes inactivate IAA-amino acid conjugates. a**, **b** Primary root lengths of 7-d-old DAO-overexpressing *Arabidopsis GFP-AtDAO1* and *GFP-AtDAO2* and rice *GFP-OsDAO* and *B. distachyon GFP-BdDAO* lines were measured. The seedlings were grown for 7 days on vertical GM plates with or without IAA-Glu-DM. Scale bar, 10 mm. The different letters represent statistical significance at *P* < 0.001 (Tukey's HSD test, *n* = 22–27). **c** Phenotypes of primary inflorescences from 35-d-old WT, *dao1-1*, and complemented lines (*GFP-AtDAO1* and *GFP-OsDAO*). At least five plants from each line were examined. Scale bar, 10 mm. This experiment was repeated three times independently.

slightly auxin-deficient phenotypes, such as short hypocotyls and fewer lateral roots, as previously reported in the Ws background[19]. The *ilr1 ill2 iar3 ill3* quadruple mutants in the Col background exhibited more severe auxin-deficient phenotypes in hypocotyl elongation and lateral root formation (Fig. 5c, d). These data are in agreement with the idea that *ILR1/ILL* genes are involved in IAA release from the IAA storage forms IAA-Asp and IAA-Glu in planta.

The *ilr1* mutation dramatically decreased the sensitivity of the *dao1-1* mutant to IAA-Glu and its methyl ester (Fig. 3d and Supplementary Fig. 17a). The *ilr1 iar3 dao1-1* and *ilr1 ill2 iar3 dao1-1* mutants showed lower sensitivity to IAA-amino acid methyl esters than did *dao1-1* mutants in lateral root formation (Supplementary Fig. 17b). In contrast, overexpression of *ILR1-GFP* in the *dao1 dao2* mutant increased its sensitivity to unmodified IAA-Asp and IAA-Glu (Supplementary Fig. 17c, d). The *ilr1 iar3 dao1-1* triple mutants completely ameliorated the high-auxin phenotypes of long root hairs, many lateral roots, and large cotyledons observed in *dao1-1* mutants (Fig. 5e and Supplementary Fig. 18a, b). The impaired fertility of the primary inflorescence in *dao1-1* was also recovered in *ilr1 dao1-1*, *ilr1 iar3 dao1-1* and *ilr1 ill2 iar3 dao1-1* mutants (Supplementary Fig. 18c). This evidence indicates that high-auxin phenotype of *dao1-1* mutants are the consequence of the IAA release from IAA-Asp and IAA-Glu by ILR1/ILLs.

**ILR1 produces oxIAA from oxIAA-Asp and oxIAA-Glu.** The *ilr1* mutants showed higher resistance to IAA-Glu-DM than IAA-Asp-DM (Supplementary Fig. 14b, c). Our kinetic experiments revealed that recombinant GST-ILR1 enzyme preferred IAA-Glu ($Kcat/Km$ $344\,M^{-1}\,s^{-1}$) to IAA-Asp ($Kcat/Km$: $7.4\,M^{-1}\,s^{-1}$) (Supplementary Fig. 19a–e). Interestingly, GST-ILR1 enzyme also

efficiently hydrolyzed oxIAA-Glu ($Kcat/Km$: $3300\,M^{-1}\,s^{-1}$) and oxIAA-Asp ($Kcat/Km$: $28\,M^{-1}\,s^{-1}$) to produce oxIAA (Fig. 6a, b and Supplementary Fig. 19f, g), implying that ILR1 hydrolyzes oxIAA-Glu and oxIAA-Asp to produce oxIAA in planta.

Consistent with ILR1 enzyme activities for hydrolyzing oxIAA conjugates, our metabolite analysis demonstrated that the levels of endogenous oxIAA and dioxIAA dramatically decreased, and whereas oxIAA-Glu greatly accumulated in the two *ilr1* mutants (Fig. 6c and Supplementary Fig. 20a). The complemented line, *pILR1::ILR1-GFP* in *ilr1*, showed the same oxIAA and oxIAA-Glu levels as WT. In the feeding experiment, the oxIAA and dioxIAA levels were elevated in WT after treatment with IAA, IAA-Asp-DM, and IAA-Glu-DM (Fig. 6c). Additionally, oxIAA-Glu levels were enormously elevated in *ilr1* plants after IAA or IAA-Glu-DM treatment (Fig. 6d and Supplementary Fig. 20). In contrast, oxIAA and dioxIAA were not produced in the *ilr1* mutant after incubation with IAA, IAA-Asp-DM, and IAA-Glu-DM (Fig. 6c, d and Supplementary Fig. 20a, b), indicating that oxIAA-Asp and oxIAA-Glu were converted to oxIAA by *ILR1*. In contrast to the significant accumulation of oxIAA-Glu in *ilr1* mutants, the oxIAA-Asp level was slightly elevated in *ilr1* mutants even after IAA and IAA-Asp-DM treatments (Fig. 6d and Supplementary Fig. 20d), suggesting that oxIAA-Asp is metabolized by other enzymes in addition to ILR1.

## Discussion

Inactivation of auxin plays a key role in maintaining auxin homeostasis in vivo, but the previously proposed molecular mechanisms governing auxin inactivation are fragmented and lack a clear picture. Several enzymes have previously been implicated in IAA inactivation by amino acid conjugation, methylation, glycosylation, and oxidation, and the IAA-

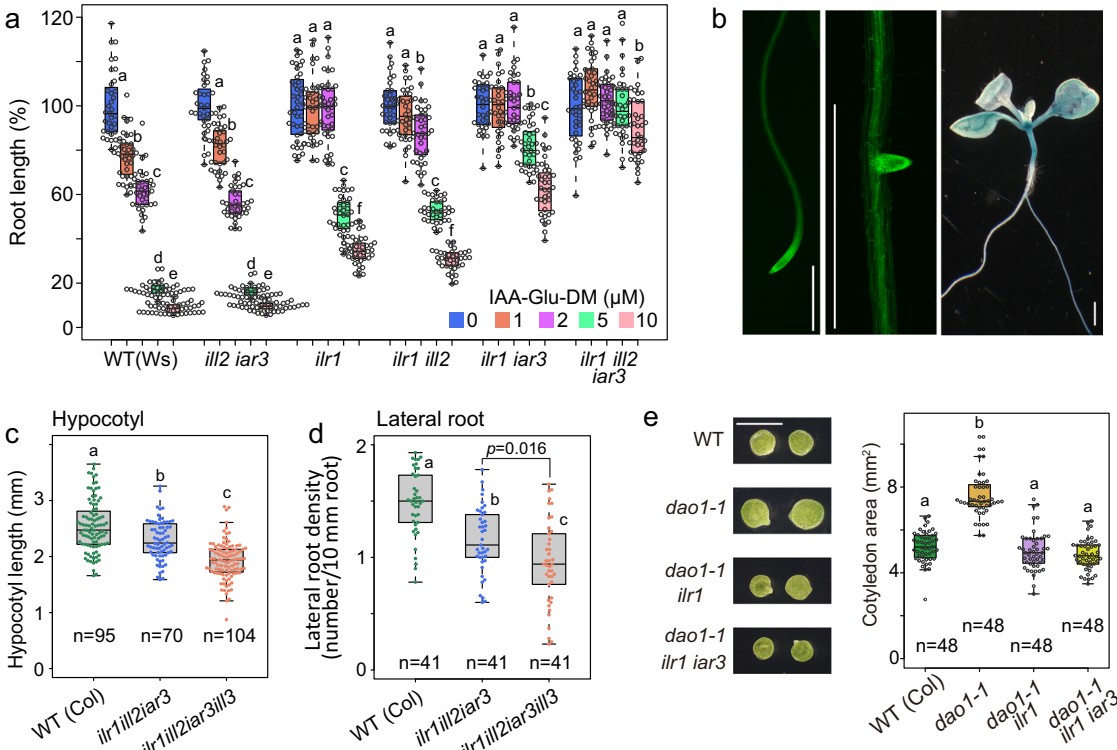

**Fig. 5 ILR1/ILL enzymes are involved in IAA release from IAA-Asp and IAA-Glu in _Arabidopsis_. a** Sensitivity of _ilr1/ill_ mutants (Ws background) to IAA-Glu-DM. Root length (6-d-old) is shown as the percentage of that in mock-treated plants (100%). The different letters represent statistical significance at $P < 0.001$ (Tukey's HSD test, $n = 34$–$42$). **b** Expression patterns of ILR1-GFP and ILR1-GUS fusion proteins under the _ILR1_ promoter. At least five plants from each line were examined. Scale bars, 1 mm. **c**, **d** Hypocotyl length and lateral root number of 8-d-old _ilr1 ill2 iar3 ill3_ quadruple mutants. The different letters represent statistical significance at $P < 0.001$ (Tukey's HSD test, b–c in Fig. 5d: $P = 0.016$). **e** Cotyledon area of _dao1-1_, _dao1-1 ilr1_, and _dao1-1 ilr1 iar3_ mutants. The plants were grown on vertical GM plates for 8 days. Scale bar, 5 mm. The different letters represent statistical significance at $P < 0.001$ (Tukey's HSD test).

inactivating enzymes appear to participate in different catabolic routes[2–4]. In this article, we established a complete pathway for oxidative inactivation of IAA, and we were able to arrange the previously described auxin-inactivating enzymes in the same metabolic pathway, thereby providing a clear picture of auxin inactivation. The GH3-ILR1-DAO pathway clearly defines the roles of three enzymes in storing auxin, reactivating stored auxin, and irreversibly deactivating auxin (Fig. 1).

Our genetic and metabolic analyses demonstrate that GH3 enzymes play a major role in auxin inactivation. IAMT1 and UGT84B1 did not redundantly compensate for GH3-mediated IAA inactivation (Fig. 2 and Supplementary Fig. 2), suggesting that IAMT1 and UGT84B1 likely play only a minor role in IAA inactivation[5,27]. DAO-mediated oxidative IAA inactivation is believed to be irreversible degradation[3,23], thereby regulating the basal level of IAA. However, we clearly demonstrate that AtDAO1 irreversibly oxidizes IAA-Asp and IAA-Glu, the IAA storage forms, to oxIAA-Asp and oxIAA-Glu, respectively, thereby modulating the levels of the storage forms in _Arabidopsis_ plants (Fig. 3 and Supplementary Fig. 4). The metabolic analysis of _ilr1_ mutants and kinetic study for ILR1 enzyme revealed that ILR1 converts oxIAA-Asp and oxIAA-Glu to oxIAA in _Arabidopsis_ plants (Fig. 6 and Supplementary Fig. 19, 20). Our feeding experiments showed that oxIAA and dioxIAA levels in _ilr1_ mutants remain unchanged even after IAA treatment (Fig. 6d and Supplementary Fig. 20b). Similarly, previous work demonstrated that oxIAA was not produced from exogenous IAA in _dao1-1_ mutants[10]. These metabolite analyses firmly established that ILR1 is essential for oxIAA production and that DAO1 did not directly

convert IAA to oxIAA. This evidence suggests that most of oxIAA is produced from IAA-amino acid conjugates by DAO1 and ILR1 enzymes, but not by enzymatic or non-enzymatic direct oxidation of IAA. Although the feeding experiment showed that [$^{13}C_6$] oxIAA is not synthesized from [$^{13}C_6$]-labeled IAA in _gh3-sept_ mutants during 20 h incubation (Fig. 2e), _gh3-sept_ mutants still accumulate endogenous oxIAA (30–40% amounts in WT plants) that were higher than the levels in the _dao1 dao2_ and _ilr1_ mutants (Fig. 2c). At a high level of IAA in _gh3-sept_ mutants, the group III GH3 enzymes, such as AtGH3.15 might produce IAA-amino acid conjugates to compensate for an impaired IAA inactivation in _gh3-sept_ mutants, thereby accumulating small amounts of oxIAA. Recent work on IAA metabolite analysis indicated that oxIAA-Asp did not accumulate in _DAO1_ knockout tobacco BY-2 cultured cells and _Arabidopsis dao1-1_ mutants, suggesting that DAO1 oxidizes IAA-Asp in planta[28].

In contrast to the previous model that IAA-Asp and IAA-Glu are not a storage form of auxin[2,4], our chemical biology study demonstrates that the two IAA conjugates actually serve as the main IAA storage forms that are readily converted back to free IAA by ILR1/ILL hydrolases in planta (Fig. 1c). IAA-Asp and IAA-Glu accumulate as major metabolites after exogenous IAA treatments[13]. Furthermore, IAA-Asp and IAA-Glu strongly accumulate in the _dao1-1_ mutant[10]. IAA-Ala and IAA-Leu have been previously thought to be IAA storage forms. However, the two conjugates have not been detected as endogenous metabolites in the WT and _dao1-1_ mutant, although _dao1-1_ is hypersensitive to exogenous IAA-Ala and IAA-Leu (Supplementary Fig. 10c, d)[29]. IAA-Ala and IAA-Leu were also not detected in rice[30]. Our in vitro

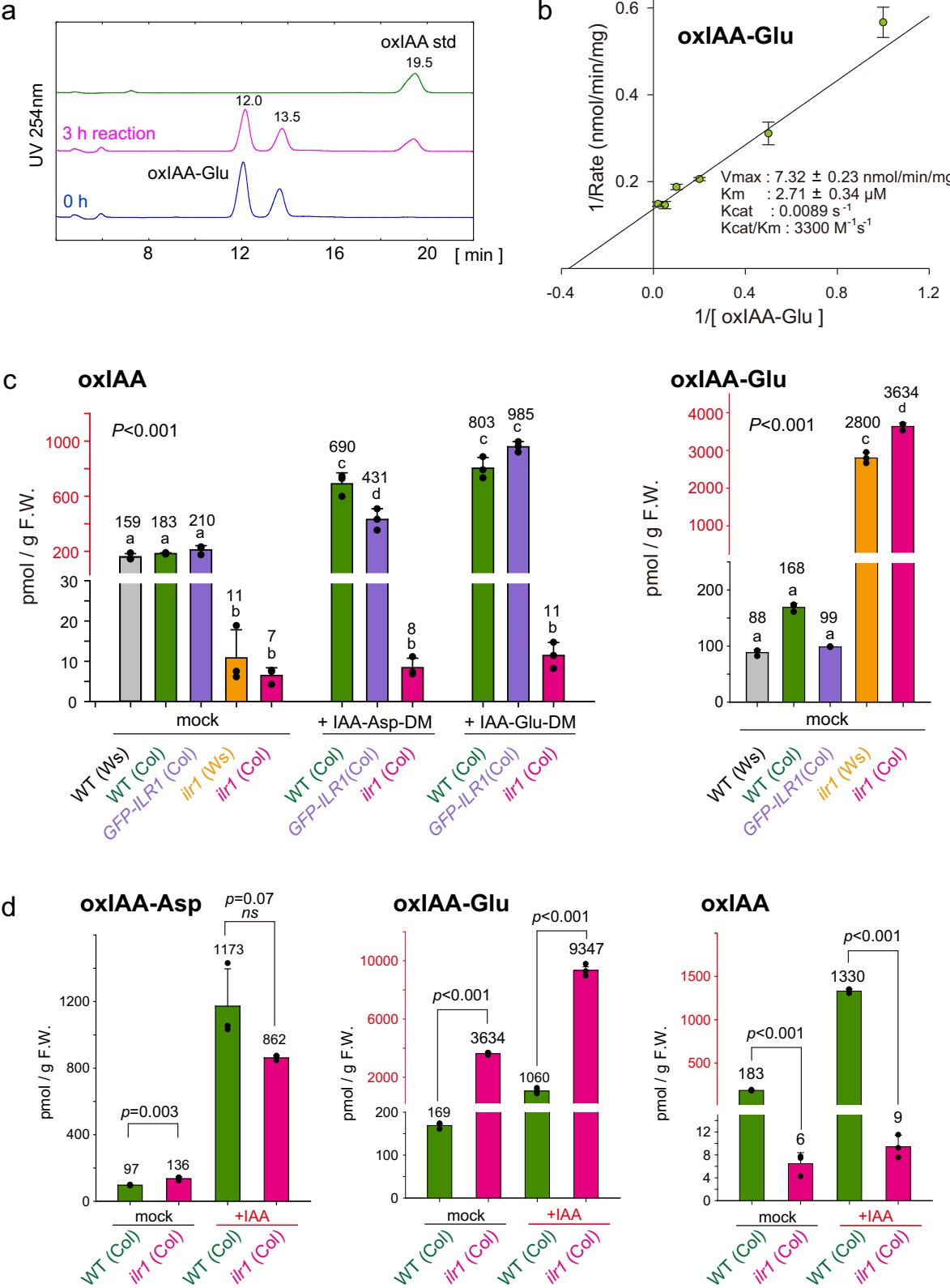

**Fig. 6 ILR1 hydrolyzed IAA amino acid conjugates to produce oxIAA. a** HPLC chromatogram of the reaction products of the recombinant GST-ILR1 enzyme. **b** Lineweaver-Burk plot and kinetics parameters of GST-ILR1 for oxIAA-Glu as the substrate ($n = 4$). **c** The endogenous levels of oxIAA and oxIAA-Glu in WT, *ilr1-1* (Ws, EMS mutant), *ilr1* (Col, SAIL_631_F01) and *pILR1::ILR1-GFP* in *ilr1* (Col) complementation lines. Seven-day-old plants were treated with 0.5 μM IAA-Asp-DM or IAA-Glu-DM for 20 h. The values shown are the means ± SD ($n = 3$). The different letters represent statistical significance at $P < 0.001$ (Tukey's HSD test). **d** The endogenous levels of oxIAA-Asp, oxIAA-Glu, and oxIAA in WT and *ilr1* (Col) seedlings. Seven-day-old plants were treated with 0.5 μM IAA for 24 h. The values shown are the means ± SD (two-tailed Student's *t* test, $n = 3$).

enzymatic assays for OsDAO demonstrate that OsDAO recognizes IAA-Asp and IAA-Glu as substrates but not IAA-Ala and IAA-Leu (Supplementary Fig. 6). Thus, IAA-Ala and IAA-Leu are unlikely to function as IAA storage forms in *Arabidopsis* and rice.

In monocots, *O. sativa* and *B. distachyon*, both IAA-Asp and IAA-Glu also function as storage forms, and the two IAA conjugates are eventually oxidized to dioxIAA via oxIAA conjugates as intermediates (Supplementary Fig. 8, 13). Notably, dioxIAA accumulates 50–200 times more than oxIAA in these monocot plants, suggesting that oxIAA is readily oxidized to dioxIAA (Supplementary Fig. 13). Similar to *B. distachyon*, the gymnosperm *Picea abies* did not accumulate oxIAA and oxIAA-glucoside but accumulated IAA-Asp after IAA application[31], suggesting that these plants might further oxidize oxIAA to dioxIAA. On the other hand, in *Arabidopsis*, UGT74D1 further converts oxIAA to oxIAA-glucoside as an abundant metabolite[32]. OxIAA displays weak inhibition on primary root growth in *Arabidopsis*[9]. These metabolic conversions of oxIAA to dioxIAA or oxIAA-glucoside may serve to minimize the adverse biological activity of oxIAA accumulated at a high level of IAA. In contrast to oxIAA-Glu accumulation, oxIAA-Asp did not accumulate in *ilr1* mutants, suggesting that *Arabidopsis* plants have another degradation pathway for oxIAA-Asp in addition to the ILR1 pathway (Supplementary Fig. 20c, d). This additional pathway would repress oxIAA accumulation from oxIAA-Asp at a high level of IAA.

Previous studies illustrated that the loss of IAA oxidation leads to the accumulation of IAA, resulting in high-auxin phenotypes of *dao1-1*[10,11,33]. We demonstrate that IAA release from IAA-Asp and IAA-Glu is the leading cause of the high-auxin phenotypes in *dao 1-1* (Fig. 5e and Supplementary Fig. 18) and that *ILR1* and *IAR3* primarily contribute to this conversion. *Arabidopsis dao1-1* mutants showed impaired fertility of the primary inflorescence, and the rice *dao* mutants were completely sterile[10–12], suggesting that ILR1/ILL and DAO-regulated auxin homeostasis is required for anther dehiscence, pollen fertility, and seed initiation. In our expression analysis, ILR1 and IAR3 were expressed in morphogenic auxin maxima (Fig. 5b and Supplementary Fig. 16). Furthermore, *ilr1 ill2 iar3 ill3* mutants showed auxin-deficient phenotypes (Fig. 5c, d). Taken together, our results indicate that *ILR1/ILL* plays a role in auxin homeostasis throughout plant development.

Our biochemical study demonstrated that ILR1 exhibited 50 times lower catalytic efficiency (Kcat/Km) for IAA-Asp than for IAA-Glu (Supplementary Fig. 19). *GH3.17* constantly converts IAA to IAA-Glu to maintain basal IAA inactivation. Alternatively, auxin-inducible *GH3* (*GH3.1–GH3.6*) produces large amounts of IAA-Asp under high levels of endogenous IAA. Given that IAA is more rapidly converted to IAA-Asp rather than to IAA-Glu at an elevated IAA level, the released IAA from IAA-Asp might be decreased when endogenous IAA level is elevated, suggesting that different role of the storage forms, IAA-Asp and IAA-Glu in IAA homeostasis in *Arabidopsis* plants

In conclusion, we present a major auxin inactivation pathway modulating auxin homeostasis. Our results demonstrate that IAA homeostasis can be maintained by the balance among three components, GH3, ILR1/ILL, and DAO, paving the way for further research seeking to elucidate the dynamics of auxin gradient formation.

## Methods

### Plant materials and growth conditions
*Arabidopsis thaliana* accession Columbia (Col) was used as the wild-type unless otherwise stated and *Arabidopsis thaliana* accession Wassilewskija (Ws) was used for Figs. 5a, 6c and Supplementary Fig. 14). Transgenic seeds of *DR5::GUS*[34], *35 S::YUC1*[35], *35 S::AtDAO1*[10], *pMDC7::IAMT1*[6], *pMDC7::GH3.6*[32], *pMDC7::YUC2*[36], *pMDC7::UGT84B1*[8] were previously described. *Arabidopsis* the *ilr1-1*, *ill2-1*, *iar3-2* single mutants, the *ilr1-1 ill2-1*, *ilr1-1 iar3-2*, *ill2-1 iar3-2* double mutants, *ilr1-1 ill2-1 iar3-2* triple mutants (Ws background)

were previously reported[19]. The *ilr1-1* and *ilr3-2* mutants are EMS mutated lines and *ill2-1* mutant is T-DNA insertion line in Ws background. *Arabidopsis thaliana* mutant lines (Col background), *ilr1* (SAIL_631_F01), *ill2* (SALK_087005C), *iar3* (SALK_022636C), *ill1* (SALK_046056C), *ill3* (SALK_092410C), *ill6* (SALK_024894C) were obtained from ABRC (Supplementary Fig. 1b). The *dao1-1* (SALK_093162), *dao2-1* (SALK_205223C)[11] and *tir1-1 afb2-3*[37], *axr1-3*[38], *slr1/iaa14*[39] mutants were previously reported. The *iamt1 c-2*, *ugt84b1-c2*, and *dao1 dao2 #31* were null mutants in the Col background. The *iamt1 c-2* and *ugt84b1-c2* CRISPR/Cas9 null mutants were previously reported[6,8]. The *gh3-sept* null mutants (homozygous in *gh3-1 2 3 4 5 6 17* and heterozygous in *gh3-9*) were used in this study as *gh3 1 2 3 4 5 6 17* octuple *null* mutants are not very fertile[21]. The *dao1 dao2 #31* double mutants were generated using CRISPR/Cas9 gene editing technology[40,41]. The target sequence for *DAO1* was CTCTCCATTGAAGATATGGGGG, and the target sequences for *DAO2* was CCCCATACACACATTGCCGAAC (PAM sites were underlined). The *dao1 dao2* mutants were genotyped with primer pair of DAO1CRP5-GT1 and DAO2CRP6-GT2, which generated a fragment of about 980 bp when the designed deletion took place. The primer pair could not amplify WT genomic DNA due to the large size of the fragment. To determine the zygosity of *dao1 dao2* mutants, we used the primer set DAO1-KO-LP + DAO1-KO-RP, which amplifies a 1247 bp fragment from WT or heterozygous DNA, but could not amplify in the homozygous mutant. The transgenic lines, *pILR1::genomeILR1-GFP/GUS*, *pILL2::genomeILL2-GFP/GUS*, *pIAR3::genomeIAR3-GFP/GUS*, *35S::GFP-AtDAO1*, *35S::GFP-OsDAO*, *35S::GFP-AtDAO2*, *35S::GFP-BdDAO*, and *35S::genomeILR1-GFP* were constructed as described in Supplementary Methods.

The *ilr1/ill* and *dao1* multiple mutants (Col background) were generated by ordinary crossing. Homozygous mutants were identified by PCR genotyping and DNA sequencing using the primers described in Supplementary Table 1. Sterilized seeds were placed on GM medium (one-half-strength Murashige and Skoog salts, 1.2% [w/v] Suc, and 0.5 g/L MES, pH 5.8, containing B5 vitamins and 0.4% [w/v] agar for horizontal culture or 0.6% [w/v] gellan gum (FUJIFILM Wako Pure Chemical, Japan) for vertical growth unless otherwise stated. The seeds were vernalized for 2–3 days at 4 °C. Seedlings were grown at 23 °C under continuous light (60–75 μmol m⁻² s⁻¹; 380–780 nm) in a growth chamber (MLR-352, Panasonic, Japan). Rice (*Oryza sativa*) Japonica Group cv. Nipponbare seeds were sterilized with 10% (v/v) bleach for 30 min and washed thoroughly with sterilized water. The seeds were germinated at 28 °C in the dark for 2 days. The germinated seeds were placed on 0.4% (w/v, sterilized water) agar containing the test compounds in a plastic culture box and cultured at 25 °C for 3 days under continuous light. *Brachypodium distachyon* Bd21 seeds were sterilized with 10% (v/v) bleach for 20 min and washed with sterilized water. The seeds were kept at 4 °C for 3 days and then cultured on 0.4% (w/v, water) agar at 24 °C for 2 days under continuous light. The germinated seeds were transferred to GM medium (0.4% w/v agar) containing the test compounds and incubated at 24 °C for another 6 days under continuous light.

### Measurements of the growth of hypocotyls, primary and lateral roots, root hairs, and cotyledon area
For primary root elongation assays, *Arabidopsis* seedlings were grown on GM plates (6 g/L gellan gum for the vertical plate culture and 4 g/L agar for horizontal plate culture) containing the indicated concentrations of chemicals for 6–8 days under continuous light at 24 °C. The lengths of the hypocotyl and primary root were recorded with a digital camera. For lateral root promotion assays, *Arabidopsis* seedlings were vertically grown for 6 days at 24 °C in continuous light on GM plates (6 g/L gellan gum). The seedlings were transferred to a horizontal GM plate (2 g/L gellan gum) containing the indicated concentrations of chemicals. The seedlings were cultivated under continuous light for another 2 days at 24 °C. The primary root length and the number of lateral roots were measured. For the measurement of root hair length, wild-type and *ilr1/ill* mutants were grown vertically on GM plates (6 g/L gellan gum) for 8 days under continuous light at 24 °C. The root hair length were recorded by a digital camera. For the measurement of cotyledon area, wild-type and mutants were grown vertically on GM plates (14 g/L agar) for 8 days under continuous light at 24 °C. The cotyledons were excised by scissors. The cotyledons were recorded with a digital camera. The image was analyzed by NIH Image J software.

### Chemical synthesis of IAA metabolites, and GH3 inhibitor
Enzyme substrate, IAA metabolites, and GH3 inhibitor (kakeimide) were synthesized as described in Supplementary Methods.

### Auxin metabolite measurements
For feeding experiments, *Arabidopsis* seedlings were grown vertically on GM agar plates (14 g/L agar) and then transferred into liquid GM media with or without chemicals. The seedlings were incubated for the indicated times under continuous light. The rice and *B. distachyon* Bd21 seedlings were incubated in 300-mL culture vials containing 40 mL of water with or without chemicals under continuous light. The roots were excised and stored in a deep freezer until use after extensive washing with distilled water.

IAA metabolites were measured by LC-MS/MS analysis[8,36]. For analysis of IAA, oxIAA, IAA-Asp, IAA-Glu, IAA-Ala, IAA-Leu, oxIAA-Asp, oxIAA-Glu and DioxIAA, frozen plant materials (30–50 mg) were prepared in three to four replicates and homogenized with zirconia beads (3 mm) in 0.3 mL of 80% acetonitrile/1% acetic acid/H₂O containing [phenyl-¹³C₆]IAA, [phenyl-¹³C₆]

oxIAA, IAA-[$^{13}C_4$, $^{15}N$]Asp, IAA-[$^{13}C_5$, $^{15}N$]Glu, [$^2H_2$]IAA-Ala, [$^2H_2$]IAA-Leu, [$^2H_2$]oxIAA-Asp, [$^2H_2$]oxIAA-Glu and [$^2H_2$]DioxIAA using a Beads Crusher µT-12 (TAITEC, Japan) for 2 min. The extracts were centrifuged at 13,000 × $g$ for 5 min at 4 °C, and the supernatant was collected. Extraction was repeated twice without internal standards. The extracts were combined and evaporated to dryness using a CVE-3000 centrifugal evaporator (EYELA, Japan). Each extract was redissolved in 1% acetic acid/$H_2O$ (1 mL), loaded onto an Oasis HLB column (1 mL; Waters, USA) and then washed with 1% acetic acid/$H_2O$ (1 mL). IAA metabolites were eluted with 80% acetonitrile/1% acetic acid/$H_2O$ (2 mL). The eluted fractions were evaporated to dryness using a CVE-3000 centrifugal evaporator. Each eluted fraction was redissolved in 1% acetic acid/$H_2O$ (1 mL) and loaded onto an Oasis WAX column (1 mL; Waters, USA)). After washing with 1% acetic acid/$H_2O$ (1 mL) and subsequent washing with 80% acetonitrile/$H_2O$ (2 mL), IAA, oxIAA and DioxIAA were eluted with 80% acetonitrile/1% acetic acid/$H_2O$ (2 mL). IAA-Asp, IAA-Glu, IAA-Ala, IAA-Leu, oxIAA-Asp and oxIAA-Glu were eluted from the column by 80% acetonitrile/0.8% formic acid/$H_2O$ (2 mL). Each fraction was evaporated to dryness using a CVE-3000 centrifugal evaporator. Then, the fractions were redissolved in 1% acetic acid/$H_2O$ (30 µL) and injected into an Agilent 6420 Triple Quad system (Agilent Technologies, USA) with a ZORBAX Eclipse XDB-C18 column (1.8 mm, 2.1 mm × 50 mm, Agilent Technologies, USA). The HPLC separation and MS/MS analysis conditions are shown in Supplementary Table 2.

**Production of recombinant AtDAO1, OsDAO1, and GST-ILR1.** Full-length cDNA clones of *AtDAO1* and *ILR1* used in this work was developed by the plant genome project of RIKEN Genomic Sciences Center (Japan)[42]. The ORF of *OsDAO* [LOC4336150, Os04g0475600] was synthesized and codon-optimized for expression in *Escherichia coli* described in Supplementary methods. The full-length ORF of *AtDAO1* was amplified by PCR with two pCold-AtDAO1 primer sets and cloned into the *BamHI* and *HindIII* sites of the pCold I vector by In-Fusion Cloning (Takara Bio, Japan). *OsDAO* was also cloned into the *KpnI* and *HindIII* sites of the pCold I vector. For AtDAO1 protein expression, *E. coli* BL21 carrying the pCold-AtDAO1 plasmid was cultured according to manufacturer's instruction (Takara Bio, Japan). For OsDAO expression, the pCold-OsDAO vector and the pG-Tf2 plasmid (Takara Bio, Japan) expressing the molecular chaperonins Tig, GroEL and GroES were cotransformed into the BL21 strain. The BL21 strain carrying pCold-OsDAO and pG-Tf2 was cultured according to the manufacturer's instruction (Takara Bio, Japan)

The cultured cells were collected and suspended in BugBuster Protein Extraction Reagent according to the manufacturer's instructions (Merck Millipore). The supernatant was collected by centrifuging at 12,000 × $g$ at 4 °C for 20 min and then applied to TALON metal affinity resin (Clontech, USA). The column was washed with Buffer I (50 mM potassium phosphate buffer, pH 7.8, 300 mM NaCl, 5 mM imidazole), and the recombinant His-AtDAO1 or His-OsDAO proteins were eluted with Buffer II (50 mM potassium phosphate buffer, pH 7.8, 300 mM NaCl, 200 mM imidazole) and dialyzed against 50 mM potassium phosphate buffer (pH 7.8). The purified His-AtDAO1 and His-OsDAO proteins (with 20% glycerol) were stored at −80 °C. The concentration of recombinant proteins was measured with the Bradford assay using BSA as a standard.

GST-ILR1 protein was expressed by a wheat germ cell-free protein expression system. The full-length ORF of *ILR1* (AT3G02875) lacking N-terminal signal sequences (23 amino acids from Met) was amplified by PrimeSTAR Max DNA Polymerase with two pEU1-ILR1 primer sets and cloned into the *BamHI* and *KpnI* sites of the pEU-E01-GST-PS-MCS-N vector (CellFree Sciences Co., Ltd. Japan) by In-Fusion Cloning. The GST-ILR1 fusion protein was synthesized by a WEPRO7240 Expression Kit (CellFree Sciences Co., Ltd. Japan) and purified with a Glutathione-Sepharose column according to the manufacturer's instructions (Cytiva, USA). The concentration of GST-ILR1 was measured with the Bradford assay.

**Enzyme activity of DAO enzymes.** For the enzyme reaction of AtDAO1 with IAA-amino acid conjugates, the reaction mixture contained 50 mM potassium phosphate buffer (pH 7.2), 0.5 mM 2-oxoglutarate, 2 µM ammonium iron(II) sulfate [$(NH_4)_2Fe(SO_4)_2 \cdot 6H_2O$], and 5 mM ascorbic acid in a final volume of 120 µL; 5–200 µM IAA-amino acid conjugates and AtDAO1 enzymes (80 ng/120 µL for the kinetic analysis of IAA-Asp, 64 ng/120 µL for the kinetic analysis of IAA-Glu, and 100–300 ng/120 µL for the other reactions in this study) were added. The enzyme reaction was carried out at 30 °C for 3 min (kinetic analysis) or for 15-120 min (the other reactions). For the enzyme reaction of OsDAO for IAA-amino acid conjugates, the reaction mixture contained 50 mM potassium phosphate buffer (pH 7.2), 0.5 mM 2-oxoglutarate, 10 µM ammonium iron(II) sulfate [$(NH_4)_2Fe(SO_4)_2 \cdot 6H_2O$], and 5 mM ascorbic acid in a final volume of 100 µL; 500 µM IAA-amino acid conjugates and OsDAO1 enzymes (800 ng/100 µL) were added. The enzyme reaction was carried out at 30 °C for 20 min. Then, 100 µL of the reaction mixture was added to 400 µL of methanol containing 50 mM phosphoric acid in a 1.5-mL microtube to terminate the reaction. After centrifugation at 12,000 × $g$ for 10 min at 2 °C, the supernatant was immediately analyzed by HPLC (EXTREMA, JASCO Japan) as described in Supplementary Methods. The oxIAA-amino acids were detected as a diastereomeric mixture in the reaction mixture. One of the diastereomers of oxIAA-amino acid was readily epimerized during sample

preparation (Supplementary Fig. 4g). For the reaction conditions for IAA, the reaction mixture contained 50 mM potassium phosphate buffer (pH 7.2), 0.5 mM 2-oxoglutarate, 0.2 mM ammonium iron(II) sulfate [$(NH_4)_2Fe(SO_4)_2 \cdot 6H_2O$], and 5 mM ascorbic acid in a final volume of 120 µL. IAA (1000 µM) and boiled or unboiled AtDAO1 enzymes (11.5 µg/120 µL) were added, and the reaction mixture was incubated in a PCR tube at 30 °C for 20 h with a thermal cycler (MJ Mini, Bio-Rad). Then, 100 µL of the reaction mixture was added to 400 µL of methanol containing 50 mM phosphoric acid in a 1.5-mL microtube to terminate the reaction. After centrifugation at 12,000 × $g$ for 10 min at 2 °C, an aliquot of the supernatant was immediately analyzed by an HPLC system (EXTREMA, JASCO Japan) as described in Supplementary Methods. For the OsDAO reaction for IAA, OsDAO enzymes (800 ng/100 µL) were added to the reaction mixture and incubated at 30 °C for 200 min. After the enzyme reaction, 100 µL of the reaction mixture was mixed with 400 µL of methanol containing 50 mM phosphoric acid in a 1.5-mL microtube to terminate the reaction. After centrifugation at 12,000 × $g$ for 10 min at 2 °C, the supernatant was immediately analyzed by an HPLC system (EXTREMA, JASCO Japan) as described in Supplementary Methods.

**Enzyme activity of GST-ILR1 enzyme.** For the GST-ILR1 enzymatic reactions for IAA-amino acids and oxIAA-amino acids, the reaction mixture consisted of 50 mM Tris-HCl buffer (pH 7.4) containing 0.5 mM dithiothreitol, 1 mM $MnCl_2$, and the substrates (50–1000 µM for IAA-Asp, 20–1000 µM for IAA-Glu, 10–200 µM for oxIAA-Asp and 1–50 µM for oxIAA-Glu) and 40–500 ng boiled or unboiled GST-ILR1 enzymes (2–10 ng/µL final conc.) in a final volume of 20 µL. The reaction mixture in the PCR tube was incubated at 32 °C for 6 h with a thermal cycler (MJ Mini, Bio-Rad). After the enzyme reaction, 20 µL of methanol containing 50 mM phosphoric acid was added to the reaction mixture to terminate the reaction. After centrifugation, 5 µL of the supernatant was immediately analyzed by an HPLC system (EXTREMA, JASCO Japan) as described in Supplementary Methods.

**Microscopy.** Photographic images were taken with a SZX16 microscope (Olympus, Japan), and fluorescent images were collected with an FV-3000 laser scanning confocal microscope (Olympus, Japan) using 488-nm light combined with 500–550-nm filters for GFP.

**Statistical analysis.** Statistical analysis was performed by One-way ANOVA Tukey's *post hoc* test and two-tailed t-test. Box plots in all figures show the median as lines in the box, the 1st to 3rd quartiles as bounds of box, whiskers with end caps extending 1.5-fold interquartile range beyond the box, and the end caps represent the minimum and maximum. Box plot overlaid with beeswarm plot was drawn with RStudio software. For LC-MS/MS analysis, three or four biological replicates in an independent experiment were analyzed and the means of the replicates were indicated.

**Reporting summary.** Further information on research design is available in the Nature Research Reporting Summary linked to this article.

## Data availability

All data are available in the main text or the supplementary materials. The clones and plant materials generated during this study are available from the corresponding authors upon request. Source data are provided with this paper.

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

## Acknowledgements

The authors thank Prof. Bonnie Bartel (Rice University) for providing the mutant seeds and helpful comments. We thank the members of the K. Hayashi and H. Kasahara laboratories for assistance in the measurement of HPLC, LC-MS/MS and phenotypic data. This work is supported by grants from the Japan Society for the Promotion of Science Grants-in-Aid for Scientific Research (JP19H03253 to K.H.; JP20K21419 and JP21H02501 to H.K.), Grant for Promotion of OUS Research Project (OUS-RP-21-4 to K.H.), and National Institutes of Health (GM114660 to Y.Z.).

## Author contributions

K.H. and K.F. conceived and designed the experiments. K.H. wrote the paper with help from K.F., H.K. and Y.Z. K.A., Y.T., K.F., Y.A., H.H., H.K., R.G., Y.H., C.G., Y.Z. and K.H. conducted the experiments and contributed to the study design. H.K., K.F. and K.H. analyzed the data.

## Competing interests

The authors declare no competing interests.
