## [Peer Review File · Nature Communications]

The main oxidative inactivation pathway of the plant hormone auxinREVIEWER COMMENTS

Reviewer #1 (Remarks to the Author):

This is an extremely important manuscript that challenges some conventional wisdoms in the area of auxin homeostasis. In particular, the authors present evidence that IAA is not directly converted to oxIAA in the plant, but rather that IAA is first conjugated to Asp or Glu, following which the conjugate is oxidised to oxIAA-Asp or oxIAA-Glu. These latter two compounds are then hydrolysed by ILR1 (in particular) to oxIAA.

This new model represents a dramatic shift in our understanding of IAA deactivation. Up until now, and particularly after the publication of three papers on IAA deactivation in PNAS in 2016, 2-oxidation and conjugation have been considered as alternative metabolic fates for IAA. Now, Hayashi et al present convincing evidence that conjugation and 2-oxidation are actually part of the same pathway. The significance of conjugation is enhanced in the new model, while 2-oxidation is still important, being the first irreversible step in IAA metabolism.

The authors' evidence is very strong, being based on the effects of mutations in the three classes of enzymes for IAA metabolism: GH3s, DAOs and ILR/ILLs.

However, the description of the evidence could be improved. The first piece of evidence that the previous model (depicted in Fig. 1b) may not be correct is that in the gh3 octuple mutant, oxIAA levels are reduced, not elevated (Fig 2c). However, the fact that oxIAA can be further oxidised to dioxIAA raises the possibility that in the octuple mutant, oxIAA is low because it is rapidly converted to dioxIAA. Therefore, at the end of line 127, we are left wondering, have the authors measured dioxIAA in the octuple mutant? In fact, they did, and the data are in Supplementary Fig. 3d. Strangely, this result is not mentioned in the manuscript. The only mention of Supplementary Fig. 3d that I found in the manuscript is in relation to dioxIAA levels in dao mutants. Therefore, the authors have omitted to mention a key piece of evidence for their model, which is that in the gh3 octuple mutant, not only is oxIAA reduced, but so too is dioxIAA.

The second main line of evidence for the authors' new model comes from the effects of mutations in the ILR1 gene. This is probably the first time that a range of IAA metabolites has been measured in ilr1 mutants. If that is the case, the authors should point it out. In the ilr1 mutants, oxIAA and di oxIAA levels were dramatically reduced, indicating that most oxIAA in the Arabidopsis plant comes from oxIAA-Asp or oxIAA-Glu, as a result of the hydrolytic action of the ILR1 hydrolases. This contrasts with the old model, in which oxIAA is derived from IAA itself.

Importantly, even after IAA treatment, oxIAA and dioxIAA levels remained low in the ilr1 mutant (Supplementary Fig. 20)

These results are perhaps the most surprising and instructive findings in the manuscript. Once again, however, the authors understate this finding, this time in the Discussion. In the section beginning on line 295, instead of concentrating on the evidence that most oxIAA comes from oxIAA-Asp and/or oxIAA-Glu, the authors quickly move onto a secondary issue, namely the possibility of other metabolic fates of oxIAA-Asp.

Furthermore, in the section beginning on line 234, the authors should more strongly highlight their evidence that recombinant ILR1 can convert IAA-Glu and IAA-Asp to IAA (Supplementary Fig 19). This is because of their controversial claim that these two amide conjugates are actually storage forms for IAA. Also relevant is their claim, in lines 77 and 78, that the ILR/ILL enzymes have never been associated with the metabolism of IAA-Asp and IAA Glu, referencing a LeClere et al paper to support that. However, this is not correct, because in fact LeClere et al did demonstrate some conversion of IAA-Asp and IAA-Glu to IAA by ILR1, similar to the conversion reported in Supplementary Fig. 19 in this manuscript.

In identifying products formed by recombinant enzymes, have the authors identified products by mass spectrometry as well as by fluorescence or UV absorbance? In Fig. 6b, the retention times of the oxIAA standard and the putative oxIAA detected as a product do not match very well. Also,

where these types of chromatograms are presented, ideally a negative control should be shown as well (if possible).

I draw the authors' attention to the fact that even in the octuple mutant, when GH3 activity is very low, some oxIAA is still present in the plant. Supplementary Fig. 3b shows that in the octuple mutant, the amide IAA conjugates are barely detectable, yet oxIAA and dioxIAA are still around 40% of the WT level. The very low levels of oxIAA and dioxIAA in *ilr1* mutants indicate that the vast majority of oxIAA is formed according to the authors' new model, so how do they explain the relatively high oxIAA and dioxIAA levels in the *gh3* octuple mutant?

Other comments:

The use of the same letter for significantly different means is very unusual. Normally significantly different means are denoted with different letters. Can the authors please review this situation?

In the "Statistical Analysis" section, it is stated that for most figures, three independent experiments were performed. So, are the means in the figures the means of values for the three experiments, or are they the means of replicates within one experiment? This needs to be clarified.

Many of the figures show means that differ greatly in magnitude. The authors have chosen to fit the axes to the large means, with the consequence that the small means are often barely visible, and the error bars are even less visible. These small means are often very important, however. I ask the authors to consider re-drawing the figures with the large means shown with breaks in them – this is the more usual method. In Supplementary Fig. 20g, the authors need to present the "expanded" version only. In this figure, what do they mean by saying that the samples were derived from the plants shown in Fig. 4a and b?

To confirm some of the conversions claimed in the manuscript, the authors could, in the future, feed isotopically labelled forms of the substrates in question and follow that up by monitoring the presence of label in the proposed product. For example, labelled IAA-asp could be fed, and if label is subsequently present in IAA, this confirms the IAA-Asp to IAA conversion in planta, assuming that not all the label is lost in the conversion. IAA-Asp can be taken up to some extent, it seems, given the DR5 signal in Fig. 3c.

Arabidopsis contains relatively large amounts of IAA-Glc (Porco et al 2016; Mellor et al, 2016). However, in the present manuscript it is claimed that the formation of IAA-Glc (by UGT84B1) is not an important deactivation step. Would the authors like to explain this apparent contradiction?

I am not convinced that the data in Supplementary Fig. 15 show an auxin-deficient phenotype of mock *ilr1 ill2 iar3 ill3* quadruple mutant plants as claimed in line 309.

The authors should select better words for clearly portraying the model shown in Fig. 5b as the previous model. Instead of describing the Fig. 5b as the "present" model, I would say the "previous" model. This could be followed by something like "(c) The new model proposed here: a reversible IAA inactivation pathway composed of"

The model shown in Supplementary Fig. 1a should also be described as the "previous model". In line 118, the authors could say: "The previous model in the literature suggests that"

Supplementary Fig. 5: I recommend replacing the word "reproduction" in the title with "release". Similar changes may be required elsewhere.

Lines 152/153: "the expectation that plant esterases would hydrolyse....." is borne out by the data, but do the authors specifically point this out anywhere in the manuscript? I recommend that the authors clearly point out somewhere in the manuscript that the activity of the IAA conjugate dimethyl esters occurs because those dimethyl esters are first converted to the amide conjugates which are then converted to IAA. Otherwise, readers might attribute some auxin-like activity to the dimethyl esters or to the conjugates themselves. These compounds are thought to be inactive without conversion to IAA.

In the paragraph starting on line 158, the authors should point out that while the dimethyl esters under discussion showed auxin activity, these compounds were much less active than IAA itself.

Is oxIAA a substrate for GH3? Did the authors test that?

Line 82, change "demonstrated" to "demonstrate"

Line 149: insert "these" after "because".

I cannot follow the logic in lines 161 to 163. What is the connection between the responses of IAA response mutants and the conversion of the IAA-amino acid diesters to free IAA?

In line 264, where we read "likely play a role", do the authors mean to write "likely do not play a role"?

I draw the authors' attention to a recent paper in *Plant Physiology*, in which evidence is presented that in conventional sample purification, indole-3-pyruvic acid can be converted to IAA, artificially inflating IAA levels (Gelinias-Marion et al, 2020). This might be relevant to future IAA determinations in the authors' laboratory. IAA levels in the *ilr1 ill2 iar3 ill3* quadruple mutant might be of interest in the future.

John Ross

Reviewer #2 (Remarks to the Author):

This is an exciting and very data-dense manuscript that overturns some of the previously postulated models of auxin conjugation and oxidation pathways in plants. While this study is conceptually remarkable and paradigm-shifting and the work is largely thoroughly executed, the manuscript was quite difficult to read and follow because of the unusually high amount of data reported (packaged into 6 main and 20 supplemental multi-panel figures, and 3 supplemental tables). In fact, a majority of statements in the text refer to several figures at once, requiring the reader to assimilate and compare multiple data points, genotypes and pharmacological treatments at once, all while reevaluating and often invalidating several of the previously established concepts in auxin biology. The manuscript felt more like two full-length articles densely packaged together into one short and very content-intensive paper, with very little space for logical connections or data interpretation. Several of the supplemental figures or panels are also referenced in the manuscript out of order, which further complicated the "digestibility" of this work. In addition, there are also some issues with how the statistical analysis is presented.

My specific comments below are presented in the chronological order.

Lines 53-54, the figure referenced (S2A) does not show the *dao1-1* mutant, so the reference to this figure with regards to the mutant's auxin over-accumulation phenotypes is unclear. Also, what is the evidence that the inducible DAO1 line shown in fig S2A indeed overexpresses DAO1? The same concern applies to other untagged overexpressors (constitutive or inducible) employed in this work.

Lines 65-67 and fig S2A, what is the evidence that the late germination or stunted growth phenotypes of GH3.6- and GH3.17-overexpressing plants are caused by auxin deficiency? I see clear gravity defects in the images, but these are not specifically called out in the text nor quantified. Also, I find it peculiar that some data figures are referenced in the introduction section of the manuscript.

Line 97, how was the KKI inhibitor designed (what was the reason to believe that this compound will block the activity of GH3s to justify its synthesis)?

Table S3 is not self-explanatory and the source of AtGH3 (recombinant?) used in the inhibitor characterization is unclear.

Fig S2B, the plants are said to be treated with 0.1 μ M IAA for 14 days. Is not IAA light sensitive? I am surprised that IAA still has an effect after two weeks of growing plants in the light.

Line 101, a reference to Fig S2C needs to be added (and removed from line 102). In the fig S2C legend, I was confused by the definition of gh3-oct with regards to gh3-9 being heterozygous (so, it is not an octuple, is it?) and the mention of gh3-9 homozygotes being insensitive to KKI (on its own or in the context of other seven GH3s being knocked out?). The mutant labeled as gh3-oct in the figure seems to be insensitive to KKI. What is the genotype (with respect to GH3-9) of the plant shown?

In Fig S2C (the first two bars) and S2E (4th and 7th bar), not all bars have a letter associated with statistical differences. Please, expand the statistical analysis to all genotypes/treatments. Herein, the same letters are said to signify the LACK of statistical difference, which is common practice.

In contrast, in Fig 2C, the same letters are said to signify the PRESENCE of statistical difference (the opposite of the Fig S2C/E bar labelling). Is that a mistake? Blue bars 2, 3, 4 for IAA lack the labeling and the genotypes that are supposed to be different based on the description in the text (e.g., the first and last blue bars in IAA) are labelled with the same letter, which I find counterintuitive and confusing. Why only these two genotypes are being compared? Likewise, in oxIAA quantification (red bars), the second red bar for oxIAA is missing a letter, and the two short bars in the middle (3 and 4) carry different letters even though they appear similar in height to one another and very different from the first and last ab bars. This is again very confusing. I would like to request that the statistical analysis is done for all values against all values, with the same letters marking non-significantly different measurements and the different letters marking the statistically different values. This should apply to all bar graphs in all figures (main and supplemental) of this paper.

Fig 2D, why is the KKI treatment alone (without IAA) not included? Again, the statistical significance of the differences being marked by the same letters does not make any sense.

Line 154 should already cite Fig S7A-B.

Line 161, *dao1* should not be referred to as an auxin biosynthesis mutant.

Fig S7D, please, include untreated controls (no auxin biosynthesis inhibitors and no IAA-Asp-DM or IAA) for all three genotypes as reference points (to be able to infer the degree of auxin-mediated complementation). What was the point of treating *sav3* with Kyn and YDF and should not have *sav3* been hypersensitive to these inhibitors? Why was *dao1* tested here and what was the expectation?

Fig S9A, what conditions were the three seedlings shown grown in, how do the three seedling images relate to the plotted data below and why only three rather than 12 seedlings (3 genotypes by 4 treatments) are displayed? What is the concentration of IAA applied in this experiment?

Fig 3D, the bar graphs appear shifted to the right with respect to the genotype labels below the x-axis.

Lines 183-184, I think saying that DAOs "restored the response to IAA-amino acid diesters in *dao1-1* mutants" is misleading. The *dao1* mutant is hypersensitive to IAA-aa diesters and the induction of DAO1 in the mutant made the plants less sensitive to this conjugate. This should not be referred to as the "restoration" of the response.

Line 198, it is redundant to refer to "rice *O. sativa*".

Fig S16A-B, it is not obvious as labeled that the GFP and GUS lines shown in the photos are in the *ilr1* and *iar3* mutant backgrounds (please, label the images accordingly). Should the last seedling in panel A be labeled as harboring the GUS reporter rather than GFP? Also, in the graphs in panels A and B, GUS and GFP are reversed; for consistency, it will be better to use the same order and color code for GFP (versus GUS) in both panels.

Line 217, given that many genes are expressed in the hypocotyls and meristematic regions, I think the patterns of expression alone do not suggest the involvement of ILR/ILL genes in auxin production. I would encourage the authors to replace "these expression patterns suggest..." with "these expression patterns are consistent with the notion that...".

Line 222, likewise, shorter hypocotyls and lateral root defects of the mutant are consistent with several different scenarios, so I'd like to request that the authors replace "these data imply that..." with "these data are in agreement with the idea that...".

Fig 5D, why do the bars are double labeled with an "a" and "b*"? Again, I find the inconsistent way to reflect statistically different data points very confusing and intuitively the same letters should mean the lack of statistically significant difference.

Fig 5E, if the authors choose to include the cotyledon images, all four genotypes should be displayed.

Fig S17A, what do the asterisks mean (typically that would mean statistically significant difference) and why was the letter format abandoned for just one panel?

Lines 228-229, this sentence does not make sense. The hypersensitivity of *dao1* is not restored but rather eliminated or abolished by *ilr1*. Also, only in fig 3D an IAA-amino acid is used (which does not test *ill2 dao1*), whereas IAA-amino acid dimethyl esters are used for the full set of mutants, so the statement is also inaccurate in that respect. The choice of the treatments/mutants is not always clear to me.

Fig S18C-D, Col control is missing but is critical as a reference point.

Fig S20A, it is unclear as labeled that the GFP-ILR1 line is in the *ilr1*(Col) background (which is stated in the legend, but not obvious from the figure itself).

Fig 6A, in the right-most graph, were the plants treated with IAA-Glu-DM? That is not marked in the figure (unlike in the left-most graph) and therefore is not at all obvious.

Line 245, a reference to Fig S20D-E is needed after the comma (as it took me a while to locate the data without that figure reference).

Line 248, should this statement reference fig S20G rather than S20F?

Line 259 and elsewhere (lines 27, 86 and 322), should the pathway be referred to as GH3-DAO-ILR1 (rather than GH3-ILR1-DAO) to refer to the postulated order of action of these enzymes in IAA inactivation?

Why are root length data sometimes presented as actual values (in mm), but in a majority of pharmacological treatments root length is expressed as relative values (% of mock treated)? This inconsistency makes it difficult to compare the data between different experiments.

Based on Fig S1B, the *ill1* T-DNA insertion resides in the promoter region. What is the evidence that this mutation disrupts *ILL1* gene expression? Likewise, the *iar3* mutant seems to be in the 3' UTR, so some evidence to suggest the lack or reduction in *IAR3* gene activity in this mutant is needed.

Is there a reason why the authors did not look at the levels of IAA-Asp and IAA-Glu in IAA-Asp-

DM- and IAA-Glu-DM-treated plants (to unequivocally demonstrate that the DM versions of these compounds are indeed hydrolyzed to IAA-Asp and IAA-Glu by endogenous esterases prior to being oxidized by DAO as the authors stipulate) or did I miss that data?

The choice of which data are displayed in the main figures versus moved to supplemental data seem to be completely random and on multiple occasions I felt that the authors chose to place critical data in the less accessible supplemental figures.

Other comments:

Figures and tables are often referenced out of order. For example, figure S3B (line 109) is mentioned before S3A (line 113); or figure S4A-D (line 132) is cited before S3C-D (line 191), etc. Table S3 (line 98) is referenced before table S1 and S2 (which are only mentioned in supplemental data), etc. Some figure panels are not referenced at all (for example, Fig S1B, Fig S4G-H, etc.).

The color of exons in Fig S1B is magenta rather than red. Please, change the legend accordingly. The reference to the magenta line in the last sentence of the legend is unclear – I am unsure what the authors are referring to. Also, the black lines that I assume represent the chromosomes are inconsistent between genes (different thickness, some filled and some are not). Please, reconcile.

The Fig S2A legend does not list the DAO1 overexpressor (please, amend). The 12 panels displaying the phenotypes of plants in plus versus minus estradiol should be aligned (6 +ER panels aligned with the respective 6 -ER panels) for easy comparison.

Fig S5I is not listed in the figure legend.

Fig S8 legend, change "Bd21 cells" to "Bd21 seeds".

Line 225, this should be "root hairs" (plural).

Supplemental Data:

Page 2, under "Plant Materials and Growth Conditions", change "state" to "stated" in the first sentence.

Page 9, Italicize *Agrobacterium tumefaciens* in the penultimate sentence.

Page 11, should the "root promotion assay" refer to a "root elongation assay"?

Reviewer #3 (Remarks to the Author):

The manuscript „Main oxidative inactivation pathway of the plant hormone auxin" by Hayashi et al presents an impressive and extensive study on how auxin is degraded in *Arabidopsis* followed by confirming similar mechanisms are in place in monocot crop plants such as rice and *Brachypodium*.

Using a thorough methodology including genetic analysis, phenotyping, metabolic analysis, kinetic assays to measure enzyme activities and modelling crystal structures they provide an impressive amount of data to demonstrate that auxin degradation is a linear pathway comprising of initial step of conjugation to amino acids, followed by oxidation and then removal of the amino acid. Oxidation by DAOs is the first irreversible step.

Importantly this novel understanding of the auxin degradation pathway is of high significance to the field. This work ties up some loose ends as some published data did not make sense with the previous model. Given the importance of auxin as signalling molecules in plants, which based on cellular concentrations regulates almost every step of development, understanding how cells remove excess auxin will significantly contribute to understanding how cells can maintain and modify their auxin levels and thereby how one molecule can regulate such a myriad of

developmental decisions.

However I do have some comments for the authors to address

1. Kakeimide presents a novel inhibitor for GH3 enzymes and I am sure this will become an important and widely used inhibitor in the field of auxin homeostasis. While the molecular structure and synthesis instructions are provided, Authors do not comment about how this was designed, tested, etc. They state this will be published elsewhere but they should provide at least the author names so this will be searchable upon publication.

2. Line 161: not IAA biosynthesis mutants have been used but wildtype was treated with IAA synthesis inhibitors Kyn + YDF Fig S7d

3. It would be useful if the authors could provide some explanation or hypothesis on why gh3oct/wt+KKI (Fig S 3) can still produce DioxIAA

4. Fig S5: Why has this not been crystallised/modelled with IAA-Glu, given the importance of this conjugate

5. Fig S9a: It's not clear with what the imaged seedlings have been treated

REVIEWER COMMENTS

Reviewer #1 (Remarks to the Author):

This is an extremely important manuscript that challenges some conventional wisdoms in the area of auxin homeostasis. In particular, the authors present evidence that IAA is not directly converted to oxIAA in the plant, but rather that IAA is first conjugated to Asp or Glu, following which the conjugate is oxidised to oxIAA-Asp or oxIAA-Glu. These latter two compounds are then hydrolysed by ILR1 (in particular) to oxIAA.

This new model represents a dramatic shift in our understanding of IAA deactivation. Up until now, and particularly after the publication of three papers on IAA deactivation in PNAS in 2016, 2-oxidation and conjugation have been considered as alternative metabolic fates for IAA. Now, Hayashi et al present convincing evidence that conjugation and 2-oxidation are actually part of the same pathway. The significance of conjugation is enhanced in the new model, while 2-oxidation is still important, being the first irreversible step in IAA metabolism.

The authors' evidence is very strong, being based on the effects of mutations in the three classes of enzymes for IAA metabolism: GH3s, DAOs and ILR/ILLs.

However, the description of the evidence could be improved. The first piece of evidence that the previous model (depicted in Fig. 1b) may not be correct is that in the *gh3* octuple mutant, oxIAA levels are reduced, not elevated (Fig 2c). However, the fact that oxIAA can be further oxidized to dioxIAA raises the possibility that in the octuple mutant, oxIAA is low because it is rapidly converted to dioxIAA. Therefore, at the end of line 127, we are left wondering, have the authors measured dioxIAA in the octuple mutant? In fact, they did, and the data are in Supplementary Fig. 3d. Strangely, this result is not mentioned in the manuscript. The only mention of Supplementary Fig. 3d that I found in the manuscript is in relation to dioxIAA levels in *dao* mutants. Therefore, the authors have omitted to mention a key piece of evidence for their model, which is that in the *gh3* octuple mutant, not only is oxIAA reduced, but so too is dioxIAA.

>Response: Thank you so much for the constructive and insightful comments on our manuscript. We have revised our manuscript accordingly. Please find below for our response to the specific comment.

First of all, according to the suggestion on the *gh3-oct* mutants by reviewer #2, we revised the description of "*gh3-oct*" mutants to "*gh3-sept*" (*gh3-1 2 3 4 5 6 17*, septuple) mutants. The *gh3-1 2 3 4 5 6 9 17* octuple null mutants are not very fertile and set very few seeds (described in the manuscript on *gh3-oct* mutants). Fig. 2a picture shows *gh3-1 2 3 4 5 6 17* (*gh3-9* heterozygous+/-) seedlings as it was genotyped. Other data using *gh3* mutants including Fig. S2c was obtained on seeds (*gh3-1 2 3 4 5 6 17*

null with *gh3-9* +/-, -/-, +/-) derived from *gh3* mutants (*gh3-1 2 3 4 5 6 17* null with *gh3-9* +/- heterozygous, genotyped). GH3.9 gene is not expressed during young seedling stage, but does function at reproductive organs (in the attached manuscript on *gh3* mutants). We could not distinguish *gh3-1 2 3 4 5 6 9 17* (*oct*) homo plants and *gh3-1 2 3 4 5 6 17* (*sept*) homo, and *gh3-1 2 3 4 5 6 17* (*gh3-9* heterozygous +/-) plants during a seedling and vegetative stage. In LC-MS/MS measurement, the genotyping of *gh3* mutants after the metabolite measurement was impossible.

According to your suggestion, we revised the following text.

(Original, line122-127)

Thus, we expected that oxIAA would be highly accumulated in *gh3-oct* mutants. Surprisingly, oxIAA was significantly decreased in *gh3-oct* mutants (Fig. 2c), and KKI repressed oxIAA production in WT and YUC2-overexpressing lines (Fig. 2d and Supplementary Fig. 3a). These results indicate that the production of oxIAA depends on the presence of GH3 enzymes and that AtDAO1 likely functions downstream of GH3 in the same pathway (Fig. 1c).

(Revised, line123)

Thus, we expected that oxIAA would be highly accumulated in *gh3-sept* mutants. Surprisingly, oxIAA was significantly decreased in *gh3-sept* mutants (Fig. 2c), and KKI repressed oxIAA production from IAA in YUC2-overexpressing and WT plants (Supplementary Fig. 3d and Fig. 2d). It is known that dioxIAA (3-hydroxy-oxIAA) is an oxidized product of oxIAA. KKI also inhibited dioxIAA production from IAA in WT (Fig. 2d). DioxIAA was significantly decreased in both the *dao1 dao2* and *gh3-sept* mutants (Supplementary Fig. 3c). Feeding experiments using ¹³C₆-labelled IAA confirmed that exogenous ¹³C₆-IAA was not converted to ¹³C₆-oxIAA and ¹³C₆-dioxIAA in *dao1 dao2* and *gh3-sept* mutants, but abundantly converted in WT plants (Fig. 2e). These results indicate that the production of oxIAA depends on the presence of GH3 enzymes and that AtDAO1 likely functions downstream of GH3 in the same pathway (Fig. 1c).

The second main line of evidence for the authors' new model comes from the effects of mutations in the ILR1 gene. This is probably the first time that a range of IAA metabolites has been measured in *ilr1* mutants. If that is the case, the authors should point it out. In the *ilr1* mutants, oxIAA and dioxIAA levels were dramatically reduced, indicating that most oxIAA in the Arabidopsis plant comes from oxIAA-Asp or oxIAA-Glu, as a result of the hydrolytic action of the ILR1 hydrolases. This contrasts with the old model, in which oxIAA is derived from IAA itself. Importantly, even after IAA treatment, oxIAA and dioxIAA levels remained low in the *ilr1* mutant (Supplementary Fig. 20) These results are perhaps the most surprising and instructive findings in the manuscript. Once again, however, the authors understate this finding, this time in the Discussion. In the section beginning on line 295, instead of concentrating on the evidence that most oxIAA comes from oxIAA-Asp and/or oxIAA-Glu, the authors quickly move onto a secondary issue, namely the possibility of other metabolic fates of oxIAA-Asp.

>Response: Thank you for valuable comments. We have revised the manuscript as indicated below.

(Original, Line 295)

We demonstrate that ILR1 converts oxIAA-Asp and oxIAA-Glu to oxIAA in *Arabidopsis* plants (Fig. 6). However, our feeding experiments indicated that oxIAA-Asp did not accumulate in *ilr1* mutants, suggesting that *Arabidopsis* plants have another degradation pathway for oxIAA-Asp in addition to the ILR1 pathway (Supplementary Fig. 20f, g). This additional pathway would degrade an excess of oxIAA-Asp at a high level of IAA and subsequently might decrease oxIAA accumulation.

(Revised, Line 289)

The metabolic analysis of *ilr1* mutants and kinetic study for ILR1 enzyme revealed that ILR1 converts oxIAA-Asp and oxIAA-Glu to oxIAA in *Arabidopsis* plants (Fig. 6 and Supplementary Fig.19 and 20). Our feeding experiments showed that oxIAA and dioxIAA level in *ilr1* mutants remain unchanged even after IAA treatment (Fig. 6d). Similarly, previous work demonstrated that oxIAA was not produced from exogenous IAA in *dao1-1* mutants¹⁰. These metabolite analyses firmly confirmed that ILR1 is essential for oxIAA production and DAO1 did not directly convert IAA to oxIAA. This evidence suggests that most of oxIAA is produced from IAA-amino acid conjugates by DAO1 and ILR1 enzymes, but not by enzymatic or non-enzymatic direct oxidation of IAA. Although the feeding experiment showed that ¹³C-oxIAA is not synthesized from ¹³C-labeled IAA in *gh3-sept* mutants during 20 h incubation (Fig. 2e), *gh3-sept* mutant still accumulates endogenous oxIAA (30-40 % amounts in WT plants) that were higher than the levels in the *dao1 dao2* and *ilr1* mutants (Fig. 2c). At a high level of IAA in *gh3-sept* mutants, the group III GH3 enzymes, such as AtGH3.15 might produce IAA-amino acid conjugates to compensate an impaired IAA inactivation in *gh3-sept* mutants, thereby accumulating small amounts of oxIAA.

We also revised Fig. 6. Supplementary Fig. S20b-d were moved to the revised Fig 6c.

Original Fig.6

Revised Fig.6.

Original supplementary Fig. 20

Revised supplementary Fig. 20.

Furthermore, in the section beginning on line 234, the authors should more strongly highlight their evidence that recombinant ILR1 can convert IAA-Glu and IAA-Asp to IAA (Supplementary Fig 19). This is because of their controversial claim that these two amide conjugates are actually storage forms for IAA.

>Response: Thank you for your valuable comments. We revised our context according to your comments.

We revised as indicated below (line 234 -)

(Original, line 234)

Our results suggested that oxIAA is produced from oxIAA-Glu or oxIAA-Asp by unknown amidohydrolases. Surprisingly, endogenous oxIAA and dioxIAA were decreased, and oxIAA-Glu was highly accumulated, in the two *ilr1* mutants (Fig. 6a and Supplementary Fig. 20a), implying that ILR1 hydrolyzes oxIAA-Glu.

(Revised, Line 237)

The *ilr1* mutants showed higher resistance to IAA-Glu-DM than IAA-Asp-DM (Supplementary Fig. 14b, c). Our kinetic experiments revealed that recombinant GST-ILR1 enzyme prefers IAA-Glu (K_{cat}/K_m 344 $M^{-1}s^{-1}$) to IAA-Asp (K_{cat}/K_m : 7.4 $M^{-1}s^{-1}$) (Supplementary Fig. 19b, d). Interestingly, GST-ILR1 enzyme also efficiently hydrolyzed oxIAA-Glu (K_{cat}/K_m : 3300 $M^{-1}s^{-1}$) and oxIAA-Asp (K_{cat}/K_m : 28 $M^{-1}s^{-1}$) to produce oxIAA (Fig. 6a, b and Supplementary Fig. 19f, g), indicating that ILR1 hydrolyzes oxIAA-Glu and oxIAA-Asp to produce oxIAA in planta.

Also relevant is their claim, in lines 77 and 78, that the ILR/ILL enzymes have never been associated with the metabolism of IAA-Asp and IAA Glu, referencing a LeClere et al paper to support that. However, this is not correct, because in fact LeClere et al did demonstrate some conversion of IAA-Asp and IAA-Glu to IAA by ILR1, similar to the conversion reported in Supplementary Fig. 19 in this manuscript.

>Response: As you suggested, LeClere et al (in Table 3, J Biol Chem, 2002, 277:20446-52) examined hydrolysis rates of twenty IAA-amino acid conjugates including IAA-Leu (56 nmol/min/mg), IAA-Asp (13 nmol/min/mg) and IAA-Glu (21 nmol/min/mg) by ILR1 enzyme. On the other hand, they also demonstrated that IAA-Asp and IAA-Glu did not show auxin-like activity in both WT and *ilr1* plants. According to your suggestion, we revised as follow.

(Original, line 77)

IAA-Leu-Resistant1 (ILR1) was initially identified as an IAA-Leu amidohydrolase from a genetic screen for mutants resistant to IAA-Leu conjugate¹⁷. The *Arabidopsis ILR1-like (ILL)* family consists of *ILR1*, *ILL1*, *ILL2*, *ILL3*, *IAR3/ILL4*, *ILL6*, and the pseudogene *ILL5*^{18, 19, 20}. ILR1/ILL enzymes convert various

IAA-amino acid conjugates to IAA *in vitro*. However, IAA-Asp and IAA-Glu, two major IAA conjugates, are recognized as nonhydrolyzable and terminal metabolites *in vivo* because IAA-Asp and IAA-Glu did not show auxin activity when added to plant growth media. Therefore, ILR1/ILL enzymes have never been associated with the metabolism of IAA-Asp and IAA-Glu.

(Revised, line 71-75)

IAA-Leu-Resistant1 (ILR1) was initially identified as an IAA-Leu amidohydrolase from a genetic screen for mutants resistant to IAA-Leu conjugate¹⁷. The *Arabidopsis* ILR1-like (ILL) family consists of ILR1, ILL1, ILL2, ILL3, IAR3/ILL4, ILL6, and the pseudogene ILL5^{18, 19, 20}. ILR1/ILL enzymes convert various IAA-amino acid conjugates to IAA *in vitro*. However, whether the ILR1/ILL enzymes are involved in metabolizing IAA-Asp and IAA-Glu in plants have not been experimentally tested because both conjugates did not show auxin-phenotypes when added to plant growth media.

In identifying products formed by recombinant enzymes, have the authors identified products by mass spectrometry as well as by fluorescence or UV absorbance? In Fig. 6b, the retention times of the oxIAA standard and the putative oxIAA detected as a product do not match very well. Also, where these types of chromatograms are presented, ideally a negative control should be shown as well (if possible).

>Response: Thank you for your crucial suggestion. All enzymatic products, oxIAA, dioxIAA, IAA-amino acids and oxIAA-amino acids were confirmed by the comparison with synthetic samples in HPLC and MS spectra. In Fig. 6b, the difference of retention times of the oxIAA standard was derived from our experimental error. We analyzed hundreds of samples to determine the kinetic parameters. Thus, we used automatic HPLC sample injectors. The HPLC retention time of oxIAA is affected by temperature and deterioration of HPLC column. Probably, we set the samples in Fig 4.A on automatic HPLC sample injector at night in winter. We turned off the air conditioning system and temperature gradually goes down and retention times might be changed.

We replaced another chromatogram of oxIAA standard. We also confirmed that heat-inactivated ILR1 did not produce IAA and oxIAA from IAA-Glu and oxIAA-Glu after incubation. But, it was not included in Figs. because the experiments were performed at different enzyme concentration and reaction volume.

Original Fig. 6b

Revised Fig.6a

I draw the authors' attention to the fact that even in the octuple mutant, when GH3 activity is very low, some oxIAA is still present in the plant. Supplementary Fig. 3b shows that in the octuple mutant, the amide IAA conjugates are barely detectable, yet oxIAA and dioxIAA are still around 40% of the WT level. The very low levels of oxIAA and dioxIAA in *ilr1* mutants indicate that the vast majority of oxIAA is formed according to the authors' new model, so how do they explain the relatively high oxIAA and dioxIAA levels in the *gh3* octuple mutant?

>Response: Thank you for your very crucial comments. I totally agree with your comments.

The oxIAA and dioxIAA level in *gh3-oct (gh3-sept)* mutants were lower than those in WT, but the oxIAA and dioxIAA level are still considerably higher than the level in *ilr1* and *dao1dao2* mutants. Our preliminary experiment found that KKI reduced only 25% of oxIAA in WT plants grown with 10 μ M KKI for 8 days. In contrast, in short-term treatment (20 h), KKI potentially inhibited oxIAA production from exogenous IAA application or in IAA overproduction *YUC2-ox* line (Fig. 2d and Fig.S3d). As you know, IAA is unstable and non-enzymatically oxidized to oxIAA. However, oxIAA level in *ilr1* mutants remains low level even after IAA treatment (Fig. 6d). These results suggest that most of oxIAA is produced by DAO1 and ILR1. We performed additional feeding experiments with ^{13}C -labeled IAA. 7-days-old WT, *dao1 dao2*, *gh3-oct (gh3-sept)* seedlings were incubated with ^{13}C -labelled IAA for 20 h. The ^{13}C -oxIAA and ^{13}C -dioxIAA were quantified by LC-MS/MS. The results clearly demonstrate that exogenous $^{13}\text{C}_6$ -IAA was not converted to $^{13}\text{C}_6$ -oxIAA and $^{13}\text{C}_6$ -dioxIAA in *dao1 dao2* and *gh3-sept* mutants, but did convert in WT plants, indicating IAA is oxidized to oxIAA via IAA-amino acid conjugates. We also examined the effects of KKI on oxIAA level in *dao1 dao2* mutants. The *dao1dao2* mutants were grown with KKI for 8 days that phenocopied high auxin phenotypes in *gh3-sept* mutants. The oxIAA level in *dao1dao2* mutants remains unchanged after KKI treatment for 8days. Thus, it is unlikely that oxIAA is directly produced from IAA by non-enzymatic oxidation of IAA. We speculate that the group III GH3 enzymes, such as AtGH3.15 having low affinity to IAA might produce IAA-amino acid conjugates at high IAA condition in *gh3-sept* and thereby storing small amounts of oxIAA. Indeed, AtGH3.15 showed the $K_{\text{cat}}/K_{\text{m}}$ value of $23 \text{ m}^{-1} \text{ s}^{-1}$ for IAA, and this $K_{\text{cat}}/K_{\text{m}}$ value was 7% of the value of AtGH3.5 ($314 \text{ m}^{-1} \text{ s}^{-1}$) (J.Biol.Chem, 293, 2018, 4277-4288, Table 2 and 3). These issues are discussed in Lien289-303 in revised text.

Revised Fig 2e

(e) [¹³C₆]IAA feeding experiment. [¹³C₆]oxIAA and [¹³C₆] dioxIAA were measured after incubation of 7-d-old WT, *dao1dao2*, and *gh3-sept* seedlings with 0.2 μM [¹³C₆] IAA for 20 h.

The seedlings were grown for 8 days with KKI and oxIAA was measured.

Fig R4.

Fig. 6d

In Fig. R4 (for the review), WT and *dao1dao2* mutants were grown for 8 days with KKI (5 μM). The WT and *dao1 dao2* mutants showed high auxin phenotype like *gh3-sept* mutants. LC-MS/MS analysis indicated that oxIAA level in *dao1dao2* remains unchanged at low level. This result suggests that oxIAA was produced from non-enzymatic oxidation from IAA. Fig.6d indicated that oxIAA level was not increased after IAA treatment in *ilr1* mutants, suggesting most of oxIAA is produced IAA-amino acid conjugates.

Other comments:

The use of the same letter for significantly different means is very unusual. Normally significantly different means are denoted with different letters. Can the authors please review this situation?

>Response: Thank you for your critical comments. The same issue has been raised by reviewer#1. This was caused by our oversight. We revised all letters representing statistical significance throughout all figures. For the statistical test, we used Tukey's multiple comparison test that evaluates among all groups in the experiment (not only vs control).

In the “Statistical Analysis” section, it is stated that for most figures, three independent experiments were performed. So, are the means in the figures the means of values for the three experiments, or are they the means of replicates within one experiment? This needs to be clarified.

>Response: Thank you for your comments.

We have revised as follow;

Statistical analysis was performed by One-way ANOVA Tukey's post hoc test. For LC-MS/MS analysis, three or four biological replicates in an independent experiment are analyzed and the means of the replicates were indicated.

Many of the figures show means that differ greatly in magnitude. The authors have chosen to fit the axes to the large means, with the consequence that the small means are often barely visible, and the error bars are even less visible. These small means are often very important, however. I ask the authors to consider re-drawing the figures with the large means shown with breaks in them – this is the more usual method. In Supplementary Fig. 20g, the authors need to present the “expanded” version only. In this figure, what do they mean by saying that the samples were derived from the plants shown in Fig. 4a and b?

> According to your suggestions, we rewrite all figures of LC-MS/MS with great differences in magnitude. Furthermore, Nature Communications requires all source data that contains all values in Figures. The source data were not required in my initial submission. In this review round, I attached source data for all

figures in our manuscript. Reader could confirm the values in all graph.

To confirm some of the conversions claimed in the manuscript, the authors could, in the future, feed isotopically labelled forms of the substrates in question and follow that up by monitoring the presence of label in the proposed product. This confirms the IAA-Asp to IAA conversion in planta, assuming that not all the label is lost in the conversion. IAA-Asp can be taken up to some extent, it seems, given the DR5 signal in Fig. 3c.

>Response: Thank you so much for your critical comments. Our evidences from the *ilr1/ill* mutants clearly indicate that ILR1/ILL is required for auxin activity of IAA-Asp/IAA-Glu-DM in planta. We also measured in vitro activity of ILR1 enzyme on IAA-Asp/Glu-DM, and found ILR1 never hydrolyze the diester forms of IAA-Asp/Glu (see Fig. R1 ILR1 did not hydrolyze IAA-Glu-dimethyl ester.). As you suggested, the direct evidence for the metabolic conversion of IAA-Asp/IAA-Glu to IAA in vivo would be crucial for our work. We performed the feeding experiment in WT and *ilr1 ill2 iar3 ill3* mutants using $^2\text{H}_2$ -labelled IAA-Asp-DM. After feeding of $^2\text{H}_2$ -IAA-Asp-DM, $^2\text{H}_2$ -labeled IAA-Asp (COOH form) and $^2\text{H}_2$ -labeled IAA were measured by LC-MS/MS. This experiment directly indicates that the metabolic conversion of IAA-Asp/IAA-Glu into IAA by ILR1/ILL enzymes in vivo. These results are shown in revised Fig. 3c and the original Fig. 3c was revised and moved to Fig. S7c.

Fig. R1 ILR1 did not hydrolyze IAA-Glu-dimethyl ester.

Revised Fig 3b

Fig 3b. *in vivo* conversion of IAA-Asp-DM to IAA. [²H₂]IAA-Asp and [²H₂]IAA were measured after incubation of 8-d-old WT and *ilr1ill2iar3ill3* seedlings with 1 μM [²H₂]IAA-Asp-DM for 6 h.

(Revised, line162)

To confirm that the IAA-amino acid dimethyl esters are metabolically converted into IAA, we analyzed the metabolites from [²H₂]-labeled IAA-Asp-dimethyl ester ([D₂]IAA-Asp-DM: ²H₂-methylene group in IAA moiety) in *Arabidopsis* plants. The unmodified [D₂]IAA-Asp was detected in WT plants after treatment with [D₂]IAA-Asp-DM (Fig. 3b). [D₂]IAA appeared in WT plants, but D₂-IAA was not detectable in *ilr1 ill2 iar3 ill3* mutants after incubation with D₂-IAA-Asp-DM (Fig. 3b), indicating that ILR1/ILL hydrolases converted IAA-amino acid to free IAA in planta. These results illustrated that plasma membrane-permeable forms of IAA conjugates, IAA-Asp-DM and IAA-Glu-dimethyl ester (IAA-Glu-DM) would be metabolically converted to the unmodified IAA-Asp and IAA-Glu, and then ILR1/ILL enzymes hydrolyze the two conjugates to release free IAA.

Arabidopsis contains relatively large amounts of IAA-Glc (Porco et al 2016; Mellor et al, 2016). However, in the present manuscript, it is claimed that the formation of IAA-Glc (by UGT84B1) is not an important deactivation step. Would the authors like to explain this apparent contradiction?

>Response: Thank you for your comments. We also published a paper for *ugt84b1* null mutants (Aoi et al, Biochem Biophys Res Commun. 2020, 532: 244-250). IAA-Glc level in *ugt84b1* null mutants were not dramatically decreased, suggesting that other UGT enzymes would be involved in IAA glycosylation. However, the redundant function among UGT84B1 and other catabolic enzymes (*IAMT1, DAO1, GH3*) in IAA catabolic pathways remained unclear. In this paper, we claimed that UGT84B would not play a major role in IAA homeostasis, however, it is possible that other IAA-glycosylation enzymes might play important role in IAA homeostasis. Thus, our description is limited to UGT84B1-mediated IAA-Glc formation. The other IAA-Glycosylation enzyme is our next important issue to be addressed.

I am not convinced that the data in Supplementary Fig. 15 show an auxin-deficient phenotype of mock *ilr1 ill2 iar3 ill3* quadruple mutant plants as claimed in line 309.

The picture of *ilr1 ill2 iar3 ill3* quadruple mutant in Supplementary Fig. 15 is 6-d-old culture (156 h culture). Indeed, at this culture stage, obvious impaired auxin phenotype would not be observed even in typical auxin mutants (see the picture indicated below). We cultured some typical auxin mutants, *tir1*, *tir1 afb2*, *axr1-3*, *iaa19/msg2*, and *taa1/sav3* in our growth condition. Some primary root has a slight defect in gravitropic response in typical auxin mutants.

ILR1/ILL would be involved in fine-tune of endogenous IAA at root tip and hypocotyls. Therefore, *ilr1/ill* mutants would be weakly impaired in auxin-related phenotypes as shown in Fig. 5c,d.

Fig S15a

Typical auxin mutants, *tir1*, *tir1 afb2*, *axr1-3*, *iaa19/msg2* and *taa1/sav3* in our growth condition.

The authors should select better words for clearly portraying the model shown in Fig. 1b as the previous model. Instead of describing the Fig. 5b as the “present” model, I would say the “previous” model. This could be followed by something like “(c) The new model proposed here: a reversible IAA inactivation pathway composed of

The model shown in Supplementary Fig. 1b should also be described as the “previous model”. In line 118, the authors could say: “The previous model in the literature suggests that

>Response: Thank you for your suggestion. According to your comments, we revised as follow;

Line 118 (Original) The **present** model suggests that *AtDAO1* is involved in basal IAA inactivation

Line118 (Revised) The **previous** model suggests that *AtDAO1* is involved in basal IAA inactivation

Fig 1 legend (Original)

Fig. 1. Structures of IAA metabolites and IAA inactivation pathways.

(a) Structures of IAA metabolites. (b) The **present** model of IAA catabolic pathways. GH3 and DAO irreversibly inactivate IAA through different pathways. (c) **Schematic representation of the reversible IAA inactivation pathway composed of GH3, ILR1, and DAO1 in Arabidopsis plants.**

Fig 1 legend (Revised)

Fig. 1. Structures of IAA metabolites and IAA inactivation pathways.

(a) Structures of IAA metabolites. (b) The **previous** model of IAA catabolic pathways. GH3 and DAO irreversibly inactivate IAA through different pathways. (c) **The new model proposed here:** the reversible IAA inactivation pathway composed of GH3, ILR1, and DAO1 in Arabidopsis plants.

Supplementary Fig. 5: I recommend replacing the word “reproduction” in the title with “release”. Similar changes may be required elsewhere.

>Response: Thank you for your suggestion. According to your comments, we revised as follow;

(Original)

Fig. 5 legend. ILR1/ILL enzymes are involved in IAA **reproduction** from IAA-Asp and IAA-Glu in Arabidopsis.

(Revised) Fig. 5 legend. ILR1/ILL enzymes are involved in IAA **release** from IAA-Asp and IAA-Glu in Arabidopsis.

>”production from IAA-Asp” was revised to “release from IAA-Asp”.

Lines 152/153: “the expectation that plant esterases would hydrolyse.....” is borne out by the data, but do

the authors specifically point this out anywhere in the manuscript? I recommend that the authors clearly point out somewhere in the manuscript that the activity of the IAA conjugate dimethyl esters occurs because those dimethyl esters are first converted to the amide conjugates which are then converted to IAA. Otherwise, readers might attribute some auxin-like activity to the dimethyl esters or to the conjugates themselves. These compounds are thought to be inactive without conversion to IAA.

I cannot follow the logic in lines 161 to 163. What is the connection between the responses of IAA response mutants and the conversion of the IAA-amino acid diesters to free IAA?

In the paragraph starting on line 158, the authors should point out that while the dimethyl esters under discussion showed auxin activity, these compounds were much less active than IAA itself.

>Response: Thank you for your comments

Lines 152/153: "the expectation that plant esterases would hydrolyse...."

Line 158: IAA-Asp-DM and IAA-Glu-DM elicited typical auxin responses, such as root growth inhibition and lateral root promotion (Fig. 3b and Supplementary Fig. 7–9).

These sentence (line 152-153 and 158) would lead to misunderstanding. Auxin activity derived from IAA-Asp/Glu dimethyl ester *in vivo* depends on two-steps metabolic activations of methyl ester hydrolysis and ILR1/ILL-mediated amido hydrolysis. Thus, we revised as follow;

(Original)

To test this hypothesis, we synthesized membrane-permeable prohormone IAA-Asp diesters with the expectation that plant esterases would hydrolyze IAA-Asp diesters to yield IAA-Asp inside the cell. We found that the diesters showed potent auxin activity in planta. On the other hand, a monoester of IAA-Asp showed weak auxin activity, perhaps due to a lower membrane permeability. We used IAA-Asp-dimethyl ester (IAA-Asp-DM) and IAA-Glu-dimethyl ester (IAA-Glu-DM) for further study (Supplementary Fig. 7a, b).

IAA-Asp-DM and IAA-Glu-DM elicited typical auxin responses, such as root growth inhibition and lateral root promotion (Fig. 3b and Supplementary Fig. 7–9). IAA-Asp-DM induced the auxin-responsive DR5 reporter (Supplementary Fig. 7c) and rescued IAA-deficient phenotypes in auxin biosynthesis mutants at 50–200 nM (Supplementary Fig. 7d). Auxin-resistant signaling mutants were insensitive to IAA-amino acid diesters (Supplementary Fig. 7e), suggesting that IAA-amino acid diesters were metabolically converted to free IAA.

(Revised, line 157-)

To test this hypothesis, we synthesized membrane-permeable prohormone IAA-Asp diesters as the pro-drug form of IAA-Asp. We found that diesters acted as pro-drugs of IAA-Asp and IAA-Glu to release

free IAA, thereby exhibiting auxin activities in planta (Supplementary Fig. 7a, b). On the other hand, a monoester of IAA-Asp showed weak auxin activity, perhaps due to a lower membrane permeability. To confirm that the IAA-amino acid dimethyl esters are metabolically converted into IAA, we analyzed the metabolites from [²H₂]-labeled IAA-Asp-dimethyl ester ([D₂]IAA-Asp-DM: ²H₂-methylene group in IAA moiety) in *Arabidopsis* plants. The unmodified [D₂]IAA-Asp was detected in WT plants after treatment with [D₂]IAA-Asp-DM (Fig. 3b). [D₂]IAA appeared in WT plants, but D₂-IAA was not detectable in *ilr1 ill2 iar3 ill3* mutants after incubation with D₂-IAA-Asp-DM (Fig. 3b), indicating that ILR1/ILL hydrolases converted IAA-amino acid to free IAA in planta. These results illustrated that plasma membrane-permeable forms of IAA conjugates, IAA-Asp-DM and IAA-Glu-dimethyl ester (IAA-Glu-DM) would be metabolically converted to the unmodified IAA-Asp and IAA-Glu, and then ILR1/ILL enzymes hydrolyze the two conjugates to release free IAA. Consequently, the released IAA elicited typical auxin responses, such as root growth inhibition and lateral root promotion (Fig. 3c and Supplementary Fig. 7–9). IAA-Asp-DM and IAA-Glu-DM induced the expression of auxin-responsive *DR5* reporter gene (Supplementary Fig. 7c). Auxin-resistant signaling mutants were insensitive to IAA-amino acid diesters (Supplementary Fig. 7d).

Line 82, change “demonstrated” to “demonstrate”

>Response: Revised as suggested.

Line 149: insert “these” after “because”.

>Response: Revised as suggested “Because these two conjugates lack auxin activity”

In line 264, where we read “likely play a role”, do the authors mean to write “likely do not play a role”?

>Response: Thank you for your comments.

IAMT1 is reported to be involved in asymmetric IAA distribution (Proc Natl Acad Sci U S A. 2018, 115(26):6864-6869.). IAA-glucoside is partially reduced in *ugt84b1* null mutant (Biochem Biophys Res Commun. 2020, 532(2):244-250.). Therefore, we described that IAMT1 and UGT84B1 likely play a role in the local and temporal inactivation of IAA. However, this sentence would misunderstand the reader. Thus, we revised as follow;

(Revised, line 284) IAMT1 and UGT84B1 did not redundantly compensate for GH3-mediated IAA inactivation (Fig. 2 and Supplementary Fig. 2), suggesting that IAMT1 and UGT84B1 likely play only a minor role in IAA inactivation.

I draw the authors’ attention to a recent paper in Plant Physiology, in which evidence is presented that in conventional sample purification, indole-3-pyruvic acid can be converted to IAA, artificially inflating IAA levels (Gelinas-Marion et al, 2020). This might be relevant to future IAA determinations in the authors’ laboratory. IAA levels in the *ilr1 ill2 iar3 ill3* quadruple mutant might be of interest in the future.

>Response: Thank you so much your valuable suggestions. We will consider this issue in our future study. In our speculation, IAA level might be decreased in *ilr1 ill2 iar3 ill3* quadruple mutants at the root tip and hypocotyls that show high expression of these genes. However, IAA homeostasis would be redundantly regulated by *GH3* and *YUC* genes. We will examine the expression of GH3 and YUC genes in quadruple mutants.

Reviewer #2 (Remarks to the Author):

This is an exciting and very data-dense manuscript that overturns some of the previously postulated models of auxin conjugation and oxidation pathways in plants. While this study is conceptually remarkable and paradigm-shifting and the work is largely thoroughly executed, the manuscript was quite difficult to read and follow because of the unusually high amount of data reported (packaged into 6 main and 20 supplemental multi-panel figures, and 3 supplemental tables). In fact, a majority of statements in the text refer to several figures at once, requiring the reader to assimilate and compare multiple data points, genotypes and pharmacological treatments at once, all while reevaluating and often invalidating several of the previously established concepts in auxin biology. The manuscript felt more like two full-length articles densely packaged together into one short and very content-intensive papers, with very little space for logical connections or data interpretation. Several of the supplemental figures or panels are also referenced in the manuscript out of order, which further complicated the “digestibility” of this work. In addition, there are also some issues with how the statistical analysis is presented.

>Response: Thank you for your positive and constructive comments and suggestions. We fully agree with this reviewer’s assessment that our manuscript is too dense to be easily followed. We have re-written many parts of the manuscript and we hope that this reviewer will find that our presentation has improved.

We revised and reorganized the order of some panels and supplemental figures. Fig. S2a-c were removed because they are not essential for our conclusion and the new Fig. S2a was added. Fig. S7d was also replaced with new results. The order of Fig. S17 and S18 were switched. In addition, Fig. 3 and Fig.6 have been revised.

My specific comments below are presented in the chronological order.

Lines 53-54, the figure referenced (S2A) does not show the *dao1-1* mutant, so the reference to this figure with regards to the mutant’s auxin over-accumulation phenotypes is unclear. Add Ref position *dao1* auxin-overexpression phenotype is reported in previous paper.

>Response: Thank you for your comments.

Fig S2a was referred in introduction section to present the phenotype on known IAA-inactivating enzymes. However, Figure S2a would not contribute much to our conclusion. Therefore, we removed Fig. S2a.

Two *Arabidopsis* DAO1 papers by Zhang and Porco (PNAS 2016) analyzed the phenotypes of DAO1 overexpression lines using the dominant *dao1-2D* line that has T-DNA insertion at 3’ UTR insertion. In Zhang’s paper, *dao1-2D* line had smaller and rounder rosette leaves and shorter inflorescences. Porco

et al also construct *35S::AtDAO1* lines and analyzed metabolite levels in both *dao1-2D* and *35S::DAO1* lines. Interestingly, IAA-Asp and IAA-Glu levels are dramatically reduced in *35S::DAO1* lines (Fig.S9 in Porco's paper). On the other hand, IAA-Asp and IAA-Glu levels are slightly decreased in *dao1-2D* line. These data indicate that *dao1-2D* seems to be a weak overexpression line. Unfortunately, the phenotype of *35S::AtDAO1* lines are not mentioned in their papers. We constructed three *DAO1* overexpression lines, *35S::AtDAO1*, *35S::GFP-AtDAO1*, ER-inducible *pMDC7::AtDAO1*. We compared the mature phenotypes of several independent lines of *35S::AtDAO1* (our line) and *35S::GFP-AtDAO1* together with *35S::AtDAO1* (S. Porco and K. Ljung line). Any obvious phenotypes like *dao1-2D* were not observed among our overexpression line and WT.

Fig. S10. in PNAS, 2016 113 (39) 11010-11015

Also, what is the evidence that the inducible *DAO1* line shown in fig S2A indeed overexpresses *DAO1*? The same concern applies to other untagged overexpressors (constitutive or inducible) employed in this work.

>Response: Thank you for your comments. We used *35S::AtDAO1* that was published by the K. Ljung lab. We also constructed *35S::AtDAO1* line (untagged) in our lab. We confirmed that both *35S::AtDAO1* lines showed high resistance to various IAA-amino acid conjugates esters. Additionally, we confirmed the expression level of *GFP-AtDAO1* is correlated to the resistance to IAA-amino acid conjugates esters (IAA-Asp-DM and IAA-Glu-DM). ER-inducible *pMDC7::AtDAO1* line in WT also showed resistance to IAA-Asp-DM after the induction of *AtDAO1* by ER. The parent line of *pMDC7::AtDAO1 / dao1-1* is *pMDC7::AtDAO1/WT* used in Fig.S2a. Other overexpression lines, *pMDC7::UGT84B*, *pMDC7::IAMT1*, and *pMDC7::GH3.6* have been published (Plant Materials and

Supplemental figure 9b

Growth Conditions section). As mentioned above, we removed Fig. S2a from the Figure S2.

Lines 65-67 and fig S2A, what is the evidence that the late germination or stunted growth phenotypes of GH3.6- and GH3.17-overexpressing plants are caused by auxin deficiency? I see clear gravity defects in the images, but these are not specifically called out in the text nor quantified. Also, I find it peculiar that some data figures are referenced in the introduction section of the manuscript.

As mentioned above, Figure S2a is not essential for our conclusions. Thus, we have removed Fig. S2a from Fig.2. We again performed the same experiment (Fig. R2) for your reference. The auxin biosynthesis inhibitors, Kyn (TAA1 inhibitor) and yucasin DF (YUC inhibitor) phenocopied *pMDC7::GH3-6*, *pMDC7::GH3-17*, *pMDC7::IAMT1*, *pMDC7::UGT84B1* lines. Exogenous 50 nM IAA could partially recover auxin deficient phenotype (stunted growth). On the other hands, three *AtDOA1* overexpressing lines showed the same response to IAA and root phenotypes as WT

Fig R2. Phenotypes of overexpression lines of *DAO1*, *GH3-6*, *GH3-17*, *IAMT1*, and *UGT84B1*. The *pMDC7* estradiol-inducible overexpression lines and *35S*-driven overexpression lines were grown vertically on GM agar plates supplemented with or without 1 μ M estradiol for 6 days.

Line 97, how was the KKI inhibitor designed (what was the reason to believe that this compound will block the activity of GH3s to justify its synthesis)? Table S3 is not self-explanatory and the source of AtGH3 (recombinant?) used in the inhibitor characterization is unclear.

>Response: Thank you for your critical comments. We are now preparing a manuscript for the design and characterization of the GH3 inhibitor, kakeimide. According to the suggestions by editor, you, and reviewer #3, we attached the preliminary draft about kakeimide and will deposit the manuscript to a preprint server soon.

A: X = CH₂, Adenosine-5'-[2-(1H-indol-3-yl)ethyl]phosphate (AIEP).
B: X = CO, Adenosine-5'-[2-(1H-indol-3-yl)acetyl]phosphate.

As you may know, the first inhibitor for GH3 acyl amino acid synthase was reported on 2012 (PLoS ONE 7(5):e37632). AIEP is an analog of IAA-AMP, an intermediate of GH3-catalyzed reaction. AIEP inhibits GH3 activity in vitro at 5-10 μ M range of K_i (KKI showed pico – nano M of K_i to GH3). We confirmed AIEP was found to be completely inactive in planta in our laboratory conditions. Another GH3 inhibitor targets the group I GH3 enzyme, GH3-11/JAR1 that function in jasmonate signaling (Nat. Chem Biol 2014 DOI: 10.1038/nchembio.1591). GH3-11/JAR1 catalyzes the conjugation of jasmonate and isoleucine to produce the active JA-Ile conjugate. We initially thought that it would be nearly impossible to find potent inhibitors that completely repress the IAA-conjugating activity of group II GH3 enzymes in vivo. Because GH3 is rapidly and highly up-regulated by its substrate IAA, suggesting that the inhibition of auxin-responsive GH3 enzymes would lead to the IAA accumulation and then the IAA-induced GH3 enzymes would overcome the inhibition by chemical inhibitors. Additionally, GH3 consists of a large gene family and using various substrates including JA, salicylic acid, IAA, and ABA, thus specificity to IAA-conjugating GH3 is highly important. Therefore, the design of in vivo effective inhibitor of group II GH3 would be highly challenging. In our draft, we determined the mode of inhibition, selectivity to GH3 enzymes, phenotypic analysis of kakeimide, GH3 inhibitor. We achieved temporal inhibition of IAA-inactivation driven by GH3. Very surprisingly, endogenous IAA level is rapidly elevated by GH3 inhibitor. The amount of endogenous IAA is almost doubled within 10 min, suggesting that turnover rate of endogenous IAA in steady state is unexpectedly fast. Thus, IAA-inactivation by GH3-DAO pathway would play a role in the regulation of rapid auxin responses. We hope that we have adequately addressed your concerns without providing additional experimental results because the paper is already data-intensive.

We reformatted the sections “Synthesis of kakeimide”, “Inhibitory activity of kakeimide on AtGH3.6” and table S3 “ K_i values of GH3 inhibitor, kakeimide on AtGH3.6”. in the supplemental methods and The preparation of recombinant GH3.6 protein was briefly documented in the methods section “Recombinant GH3.6 protein was purified by TALON metal affinity resin from the culture lysate of *E. coli* BL21 harboring pCold-GH3.6.”

Fig S2B, the plants are said to be treated with 0.1 μ M IAA for 14 days. Is not IAA light sensitive? I am

surprised that IAA still has an effect after two weeks of growing plants in the light.

>Response: Thank you for your critical comments. We fully agree with your comments. I removed Fig. S2B from our manuscript. As you pointed out, IAA is unstable compound. The cultivation time in Fig S2B would not be appropriate for experimental condition. To assess the effects of kakeimide (KKI) on IAA inactivation *in vivo*, we added new experiments in Fig S2. Revised Fig. S2b shows *DR5::GUS* expression treated with KKI and/or IAA for 6 hours. Revised Fig. S2a indicated short time effects of KKI on IAA inactivation *in vivo*.

Original Fig. S2a and S2b.

(Original, line 98-102)

WT seedlings treated with KKI phenocopied *gh3-oct* mutants, and cotreatment with IAA and KKI synergistically enhanced the high-auxin phenotypes of WT (Fig. 2a and Supplementary Fig. 2b). The *gh3-oct* mutant was insensitive to KKI. Additionally, KKI enhanced the high-auxin phenotypes in the auxin overproducing *35S::YUC1* lines (Supplementary Fig. 2c, d).

Revised Fig. S2 (Fig S2b: new experimental data)

(Revised, line 99-104)

WT seedlings treated with KKI phenocopied *gh3-sept* mutants and the *gh3-sept* mutants were insensitive to KKI treatment (Fig. 2a and Supplementary Fig. 2a). Additionally, co-treatment with IAA and KKI synergistically enhanced auxin response in the auxin-inducible *DR5::GUS* reporter line (Supplementary Fig. 2b). KKI enhanced the high-auxin phenotypes in the *35S::YUC1* lines which overproduce auxin (Supplementary Fig. 2c), suggesting that KKI elevated endogenous IAA levels by inhibiting group II GH3 activities.

Line 101, a reference to Fig S2C needs to be added (and removed from line 102). In the fig S2C legend, I was confused by the definition of *gh3-oct* with regards to *gh3-9* being heterozygous (so, it is not an octuple, is it?) and the mention of *gh3-9* homozygotes being insensitive to KKI (on its own or in the context of other seven GH3s being knocked out?). The mutant labeled as *gh3-oct* in the figure seems to be insensitive to KKI. What is the genotype (with respect to GH3-9) of the plant shown?

>Response: Thank you for your critical comments on *gh3-oct* mutants. The *gh3-1 2 3 4 5 6 9 17* octuple null mutants set very few seeds (described in the attached manuscript on *gh3-oct* mutants). We maintained the mutants as *gh3-1 2 3 4 5 6 17 9^{+/-}*. Fig. 2a picture shows *gh3-1 2 3 4 5 6 17 (gh3-9 heterozygous+/-)* seedlings as it was genotyped. Other data using *gh3* mutants including Fig. S2c was obtained on seeds (*gh3-1 2 3 4 5 6 17* null together with *gh3-9 +/-, -/-, +/+*) derived from *gh3* mutants (*gh3-1 2 3 4 5 6 17* null with *gh3-9 +/-* heterozygous, genotyped). GH3.9 gene is not expressed during young seedling stage, but does function at reproductive organs. Thus, we could not distinguish *gh3-1 2 3 4 5 6 9 17* (octuple) homo plants and *gh3-1 2 3 4 5 6 17* (septuple) homo, and *gh3-1 2 3 4 5 6 17 (gh3-9 heterozygous+/-)* plants during the seedlings and vegetative stage (described in the attached manuscript on *gh3-oct* mutants). In LC-MS/MS measurement, the genotyping of *gh3* mutants after the metabolite measurement was almost impossible. According to your suggestion, we revised the description of “*gh3-oct*” mutants to read “*gh3-sept*” mutants.

Additionally, *gh3-sept* was mentioned in the Plant Materials and Growth Conditions section.

“The *gh3-sept* null mutants (homozygous in *gh3-1 2 3 4 5 6 17* and heterozygous in *gh3-9*) were used in this study as *gh3-1 2 3 4 5 6 9 17* octuple null mutants are sterile.”

In Fig S2C (the first two bars) and S2E (4th and 7th bar), not all bars have a letter associated with statistical differences. Please, expand the statistical analysis to all genotypes/treatments. Herein, the same letters are said to signify the LACK of statistical difference, which is common practice.

In contrast, in Fig 2C, the same letters are said to signify the PRESENCE of statistical difference (the opposite of the Fig S2C/E bar labelling). Is that a mistake? Blue bars 2, 3, 4 for IAA lack the labeling and the genotypes that are supposed to be different based on the description in the text (e.g., the first and last blue bars in IAA) are labelled with the same letter, which I find counterintuitive and confusing. Why only

these two genotypes are being compared? Likewise, in oxIAA quantification (red bars), the second red bar for oxIAA is missing a letter, and the two short bars in the middle (3 and 4) carry different letters even though they appear similar in height to one another and very different from the first and last ab bars. This is again very confusing. I would like to request that the statistical analysis is done for all values against all values, with the same letters marking non-significantly different measurements and the different letters marking the statistically different values. This should apply to all bar graphs in all figures (main and supplemental) of this paper.

>Response: Thank you for your critical comments. The same issue has been raised by reviewer#1. This is caused by our oversight. We revised all letters representing statistical significance throughout all figures. For the statistical test, we used Tukey's multiple comparison test that evaluate among all groups in the experiment (not only vs control by t-test.).

Fig 2D, why is the KKI treatment alone (without IAA) not included? Again, the statistical significance of the differences being marked by the same letters does not make any sense.

>Response: Thank you for your comments. In Fig.2D (20 h incubation), essentially same data for the KKI treatment alone (without IAA) are shown in Fig. S3d (36h incubation, WT (Col) treated with KKI alone). IAA, oxIAA, IAA-Asp, IAA-Glu in KKI-treated WT (Col) plants were indicated in Fig. S 3d.

We have revised letters representing statistical significance throughout figures and supplemental figures.

Revised Fig. 2e

Revised Fig. S3d

Line 154 should already cite Fig S7A-B.

>Response: revised as suggested.

Line 161, dao1 should not be referred to as an auxin biosynthesis mutant.

>Response: Lien161 was removed as original Supplementary Fig. 7d was exchanged.

IAA-Asp-DM induced the auxin-responsive *DR5* reporter (Supplementary Fig. 7c) and rescued IAA-deficient phenotypes in auxin biosynthesis mutants at 50–200 nM (Supplementary Fig. 7d).

Fig S7D, please, include untreated controls (no auxin biosynthesis inhibitors and no IAA-Asp-DM or IAA) for all three genotypes as reference points (to be able to infer the degree of auxin-mediated complementation). What was the point of treating sav3 with Kyn and YDF and should not have sav3 been hypersensitive to these inhibitors? Why was dao1 tested here and what was the expectation?

Is there a reason why the authors did not look at the levels of IAA-Asp and IAA-Glu in IAA-Asp-DM- and IAA-Glu-DM-treated plants (to unequivocally demonstrate that the DM versions of these compounds are indeed hydrolyzed to IAA-Asp and IAA-Glu by endogenous esterases prior to being oxidized by DAO as the authors stipulate) or did I miss that data?

>Response: Thank you for your valuable comments, Fig S7d showed that IAA released from IAA-Asp-DM can restore auxin deficient phenotypes in Arabidopsis plants. As you suggested original Fig S7d would not provide crucial data to support our conclusion and confuse the reader. Taken together with the comments by reviewer #1 and you, we replaced Fig. S7d to new Fig. S7c (DR5 reporter assay). The feeding experiment using D2-labeled IAA-Asp-DM was added to Fig. 3c. The feeding experiment clearly demonstrated the metabolic conversion of IAA-amino acid conjugate diesters to IAA-amino acids and

then to free IAA.

Original Fig. S7

Revised Fig. S7

(to unequivocally demonstrate that the DM versions of these compounds are indeed hydrolyzed to IAA-Asp and IAA-Glu by endogenous esterases prior to being oxidized by DAO as the authors stipulate) or did I miss that data?

We confirmed that recombinant DAO1 and ILR1 enzymes never recognized the diester forms of conjugate as the substrate (Fig S4e and Fig. R3 indicated below), and the *ilr1 ill2 iar3 ill3* mutants did not show high auxin phenotypes in the presence of IAA-Asp-DM and IAA-Glu-DM. Therefore, we speculated

that IAA-Asp/Glu-DM is metabolically converted to IAA via IAA-Asp/Glu (COOH form) as the storage form. In the previous version of our manuscript, we did not include the LC-MS/MS data supporting the metabolic conversion of IAA-Asp-DM to IAA via IAA-Asp as intermediate. Because the addition of IAA-Asp/Glu-DM did not elevate endogenous IAA levels in WT (see Fig. S12b). The released IAA would be rapidly converted again to IAA-Asp by GH3. Thus, ^2H -labelled IAA-Asp/Glu-DM are required for this feeding experiment to distinguish endogenous IAA and the released IAA. We used IAA-Asp [^{13}C , ^{15}N -Asp] as internal standard in this study. This labeled IAA-Asp release unlabeled free IAA in plants. Thus, IAA-Asp [^{13}C , ^{15}N -Asp] diester could not be used for the feeding experiments and new isotope-labeled IAA-Asp-DM (labeled in IAA moiety), such as IAA-[α $^2\text{H}_2$]-Asp-DM are required for the feeding experiments.

As you and reviewer#1 pointed out, we realized that such data is important and thus we added feeding experiments using ^2H -labeled IAA-Asp-DM in WT and *ilr1/ill* mutants to confirm the metabolic conversion of the conjugate diesters. (in revised Fig. 3b, and original Fig. 3c was revised and moved to Fig. S7c.)

Fig. R3 ILR1 hydrolyzed IAA-Glu-COOH, but ILR1 did not hydrolyze IAA-Glu-DM ester.

Fig S12b

Revised Fig 3b

(Fig. 3b) in vivo conversion of IAA-Asp-DM to IAA. [²H₂]IAA-Asp and [²H₂]IAA were measured after incubation of 8-d-old WT and *ilr1ill2iar3ill3* seedlings with 1 μM [²H₂]IAA-Asp-DM for 6 h.

(Revised, line162)

To confirm that the IAA-amino acid dimethyl esters are metabolically converted into IAA, we analyzed the metabolites from [²H₂]-labeled IAA-Asp-dimethyl ester ([D₂]IAA-Asp-DM: ²H₂-methylene group in IAA moiety) in *Arabidopsis* plants. The unmodified [D₂]IAA-Asp was detected in WT plants after treatment with [D₂]IAA-Asp-DM (Fig. 3b). [D₂]IAA appeared in WT plants, but D₂-IAA was not detectable in *ilr1 ill2 iar3 ill3* mutants after incubation with D₂-IAA-Asp-DM (Fig. 3b), indicating that ILR1/ILL hydrolases converted IAA-amino acid to free IAA in planta. These results illustrated that plasma membrane-permeable forms of IAA conjugates, IAA-Asp-DM and IAA-Glu-dimethyl ester (IAA-Glu-DM) would be metabolically converted to the unmodified IAA-Asp and IAA-Glu, and then ILR1/ILL enzymes hydrolyze the two conjugates to release free IAA.

Fig S9A, what conditions were the three seedlings shown grown in, how do the three seedling images relate to the plotted data below and why only three rather than 12 seedlings (3 genotypes by 4 treatments) are displayed? What is the concentration of IAA applied in this experiment?

>Response: Thank you for your suggestions. Same questions were also raised by reviewer #3. We have revised as indicated below.

Fig 3D, the bar graphs appear shifted to the right with respect to the genotype labels below the x-axis.

>Response: Thank you for your suggestion. We revised as you suggested.

Lines 183-184, I think saying that DAOs “restored the response to IAA-amino acid diesters in *dao1-1* mutants” is misleading. The *dao1* mutant is hypersensitive to IAA-aa diesters and the induction of DAO1 in the mutant made the plants less sensitive to this conjugate. This should not be referred to as the “restoration” of the response.

>Response: Thank you for your comments. We revised as follow:

Overexpression of native and GFP-fused *O. sativa* OsDAO and *B. distachyon* BdDAO conferred insensitivity to IAA-Glu-DM in WT and restored the response to IAA-amino acid diesters in *dao1-1* mutants (Fig. 4a, b and Supplementary Fig. 11a).

(Revised, line195-186)

Overexpression of native and GFP-fused *O. sativa* OsDAO and *B. distachyon* BdDAO decreased the sensitivity to IAA-amino acid diesters in both WT (Fig. 4a, b) and *dao1-1* mutants (Supplementary Fig.

11a).

Line 198, it is redundant to refer to “rice *O. sativa*”.

>Response: both rice *O. sativa* and *B. distachyon* seedlings > both *O. sativa* and *B. distachyon* seedlings

Fig S16A-B, it is not obvious as labelled that the GFP and GUS lines shown in the photos are in the *ilr1* and *iar3* mutant backgrounds (please, label the images accordingly). Should the last seedling in panel A be labeled as harboring the GUS reporter rather than GFP? Also, in the graphs in panels A and B, GUS and GFP are reversed; for consistency, it will be better to use the same order and color code for GFP (versus GUS) in both panels.

>Response: Thank you so much for your suggestions. These were errors. We have pasted wrong text. We revised as following Figures.

Line 217, given that many genes are expressed in the hypocotyls and meristematic regions, I think the patterns of expression alone do not suggest the involvement of ILR/ILL genes in auxin production. I would encourage the authors to replace “these expression patterns suggest...” with “these expression patterns are consistent with the notion that...”.

Original: These expression patterns suggest that the *ILR1/ILL* genes participate in IAA production from IAA-Asp and IAA-Glu.

(Revised, line 229): These expression patterns are consistent with the notion that the *ILR1/ILL* genes participate in IAA release from IAA-Asp and IAA-Glu.

Line 222, likewise, shorter hypocotyls and lateral root defects of the mutant are consistent with several different scenarios, so I'd like to request that the authors replace “these data imply that...” with “these data are in agreement with the idea that...”.

>Response: Revised according to the suggestion.

(Original)

>These data imply that *ILR1/ILL* genes are involved in IAA production from the IAA storage forms IAA-Asp and IAA-Glu in planta.

(Revised, line 234)

>These data are in agreement with the idea that *ILR1/ILL* genes are involved in IAA release from the IAA storage forms IAA-Asp and IAA-Glu in planta.

Fig 5D, why do the bars are double labeled with an “a” and “b*”? Again, I find the inconsistent way to reflect statistically different data points very confusing and intuitively the same letters should mean the lack of statistically significant difference.

>Response: Thank you again for your critical comments. Same issue has raised by reviewer#1. We revised the letters representing statistical significance throughout all figures.

Fig 5E, if the authors choose to include the cotyledon images, all four genotypes should be displayed.

>Response: Thank you for your comments. We revised Fig. 5e according to your comments.

Fig S17A, what do the asterisks mean (typically that would mean statistically significant difference) and

why was the letter format abandoned for just one panel?

>Response: Thank you again for your critical comments. We removed the asterisks and revised the letters representing statistical significance throughout all figures according to your comments.

Lines 228-229, this sentence does not make sense. The hypersensitivity of *dao1* is not restored but rather eliminated or abolished by *ilr1*. Also, only in fig 3D an IAA-amino acid is used (which does not test *ilr2 dao1*), whereas IAA-amino acid dimethyl esters are used for the full set of mutants, so the statement is also inaccurate in that respect. The choice of the treatments/mutants is not always clear to me.

>Response: Thank you for your comments. Fig. 18a showed the response of *ilr1/dao1*, *ilr2/dao1* and *iar3/dao1* to IAA-Glu-DM. Among *ilr1*, *ilr2* and *iar3* single mutants, our results (Fig. S14c) indicated that *ilr2* single mutation did not confer the resistance to IAA-Glu esters same as *iar3*. We think *ilr2 dao1* response to IAA-Glu is not essential for Fig. 3d.

WT roots did not show sensitivity to the unmodified IAA-Asp and IAA-Glu as previously reported. We examined the effects of unmodified IAA-Asp and IAA-Glu on *dao1* mutant and *35S::ILR1* in *dao1* mutation. (Fig. 3d and Fig. S17c, d)

According to your suggestion, we revised the context (line 224-231) and the order of Figure S17 and S18 was exchanged.

(Original)

The *ilr1 iar3 dao1-1* triple mutants completely ameliorated the high-auxin phenotypes of long root hair, many lateral roots, and large cotyledons observed in *dao1-1* mutants (Fig. 5e and Supplementary Fig. 17a, b). The impaired fertility of the primary inflorescence in *dao1-1* was also recovered in *ilr1 dao1-1*, *ilr1 iar3 dao1-1* and *ilr1 ill2 iar3 dao1-1* mutants (Supplementary Fig. 17c). Moreover, the hypersensitivity of the *dao1-1* mutant to IAA-amino acids was restored by *ilr1* but not *ill2* or *iar3* (Fig. 3d and Supplementary Fig. 18a, b). In contrast, overexpression of *ILR1-GFP* in the *dao1 dao2* mutant increased its sensitivity to unmodified IAA-Asp and IAA-Glu (Supplementary Fig. 18c, d).

(Revised, line 237-)

The *ilr1* mutation dramatically decreased the sensitivity of the *dao1-1* mutant to IAA-Glu and its methyl ester (Fig. 3d and Supplementary Fig. 17a). The *ilr1 iar3 dao1-1* and *ilr1 ill2 iar3 dao1-1* mutants showed lower sensitivity to IAA-amino acid methyl esters than *dao1-1* mutants in lateral root formation (Supplementary Fig. 17b). In contrast, overexpression of *ILR1-GFP* in the *dao1 dao2* mutant increased its sensitivity to unmodified IAA-Asp and IAA-Glu (Supplementary Fig. 17c, d). The *ilr1 iar3 dao1-1* triple mutants completely ameliorated the high-auxin phenotypes of long root hairs, many lateral roots, and large cotyledons observed in *dao1-1* mutants (Fig. 5e and Supplementary Fig. 18a, b). The impaired fertility of the primary inflorescence in *dao1-1* was also recovered in *ilr1 dao1-1*, *ilr1 iar3 dao1-1* and *ilr1 ill2 iar3 dao1-1* mutants (Supplementary Fig. 18c). This evidence indicates that high-auxin phenotype of *dao1-1* mutants would attribute to IAA release from IAA-Asp and IAA-Glu by ILR1/ILLs.

Fig S18C-D, Col control is missing but is critical as a reference point.

>Response: Thank you for your comments. In this experiment, 5 μ M IAA-Asp, 10 μ M IAA-Glu, 0.1 μ M IAA-Asp-DM, and 0.5 μ M IAA-Glu-DM were used for the treatment (Fig. S17c, d in revised version). The primary root elongation in WT and *dao1-1* mutants treated with IAA-Asp-DM, IAA-Glu-DM and IAA-Glu were quantified in Fig. 3d and S9.

Fig. S18c, d (revised Fig S17, c,d)

Fig S20A, it is unclear as labeled that the GFP-ILR1 line is in the *ilr1* (Col) background (which is stated in the legend, but not obvious from the figure itself).

>Response: Thank you for your comments. We revised Fig. S20a,f,g and also Fig.6 cd. The label “GFP-ILR1” was revised to “GFP-ILR1 (Col)”.

Fig 6A, in the right-most graph, were the plants treated with IAA-Glu-DM? That is not marked in the figure (unlike in the left-most graph) and therefore is not at all obvious.

>Response: Thank you for your comments. We revised Fig. 6a, right panel. We added “mock” label same as left panel.

Line 245, a reference to Fig S20D-E is needed after the comma (as it took me a while to locate the data without that figure reference).

> Revised as suggested.

Line 248, should this statement reference fig S20G rather than S20F?

> Revised as suggested.

Line 259 and elsewhere (lines 27, 86 and 322), should the pathway be referred to as GH3-DAO-ILR1 (rather than GH3-ILR1-DAO) to refer to the postulated order of action of these enzymes in IAA inactivation?

>Response: Thank you for your suggestion. Indeed, IAA was sequentially inactivated by GH3, DAO1 and ILR1.

In the different view, the irreversible inactivation of IAA is achieved by DAO1. The GH3, ILR1 and DAO coordinately regulate the IAA homeostasis as IAA-amino acid is a key storage form. Thus we would like to designate new pathway as GH3-ILR1-DAO pathway.

Why are root length data sometimes presented as actual values (in mm), but in a majority of pharmacological treatments root length is expressed as relative values (% of mock treated)? This inconsistency makes it difficult to compare the data between different experiments.

>Response: Thank you for your comments. For the pharmacological treatments, many mutant root lengths were measured and spastically analyzed. In the multiple comparison test such as Tukey's HSD test, statistical difference between two means are strict more than *t*-test when number of groups (mutant

type) are increased. The relative values (% of mock treated) would be appropriate for the comparison of these analysis to examine the effects of hormone in dose-dependent manner. When the means from two - three plants in mock treatment are not significantly different, we indicated actual root length. In this review round, we have submitted source data that contains root length (mm) and relative values (%).

Based on Fig S1B, the *ill1* T-DNA insertion resides in the promoter region. What is the evidence that this mutation disrupts *ILL1* gene expression? Likewise, the *iar3* mutant seems to be in the 3' UTR, so some evidence to suggest the lack or reduction in *IAR3* gene activity in this mutant is needed.

>Response: Thank you for your comments. T-DNA was inserted at 3'-end within ORF in *iar3* (SALK_022636C) mutant, not in 3' UTR region. We confirmed that *iar3* (SALK_022636C) and another *iar3* T-DNA line (SALK069047) showed same insensitivity to IAA-Ala as *iar3* (Ws background, published EMS line from Bartel lab). These results indicated that *iar3* (SALK_022636C) line is loss of functional mutants same as *iar3* (Ws background, EMS line). Actually, in Fig.S 16b, the insensitivity of *iar3* T-DNA line (SALK069047) mutants to IAA-Ala ester was return to that of WT by complementation of *promoter_IAR3::IAR3-GUS* fragment.

We also confirmed *ILL1* expression and *ILL1* was down-regulated. As shown in Fig S1b, *ILL1* is located near *ILL2*. Indeed, the null KO *ill1 ill2* mutant with CRISPR-Cas would be ideal mutant. We think that this would be future work.

The choice of which data are displayed in the main figures versus moved to supplemental data seem to be completely random and on multiple occasions I felt that the authors chose to place critical data in the less accessible supplemental figures.

>Response: Thank you for your comments. To improve readability, we have moved Fig. S3e to Fig. 2e, and Fig.3c (*DR5::GUS* response) was moved to Fig. S7 and then added new data (Fig. 3c). We exchanged the order of Fig. S17 and S18.

Original Fig. 2

revised Fig. 2

original Fig. 3

revised Fig. 3

Other comments:

Figures and tables are often referenced out of order. For example, figure S3B (line 109) is mentioned before S3A (line 113); or figure S4A-D (line 132) is cited before S3C-D (line 191), etc. Table S3 (line 98) is referenced before table S1 and S2 (which are only mentioned in supplemental data), etc. Some figure panels are not referenced at all (for example, Fig S1B, Fig S4G-H, etc.).

>Response: Thank you for your comments. We carefully revised the order of reference of all figure. Fig S1B, Fig S4G-H are cited in main text and method section.

The color of exons in Fig S1B is magenta rather than red. Please, change the legend accordingly. The reference to the magenta line in the last sentence of the legend is unclear – I am unsure what the authors are referring to. Also, the black lines that I assume represent the chromosomes are inconsistent between genes (different thickness, some filled and some are not). Please, reconcile.

>Response: Thank you for your comments. We revised Fig.S1B according to your suggestions.

The Fig S2A legend does not list the DAO1 overexpressor (please, amend). The 12 panels displaying the phenotypes of plants in plus versus minus estradiol should be aligned (6 +ER panels aligned with the respective 6 -ER panels) for easy comparison.

>Response: As mentioned above, Fig. S2A was removed.

Fig S51 is not listed in the figure legend.

>Revised as suggested.

Fig S8 legend, change “Bd21 cells” to “Bd21 seeds”.

>Revised as suggested.

Line 225, this should be “root hairs” (plural).

>Revised as suggested.

Supplemental Data:

Page 2, under “Plant Materials and Growth Conditions”, change “state” to “stated” in the first sentence.

>Revised as suggested.

Page 9, Italicize *Agrobacterium tumefaciens* in the penultimate sentence.

>Revised as suggested.

Page 11, should the “root promotion assay” refer to a “root elongation assay”?

>“For lateral root promotion assays” is correct.

Reviewer #3 (Remarks to the Author):

The manuscript „Main oxidative inactivation pathway of the plant hormone auxin” by Hayashi et al presents an impressive and extensive study on how auxin is degraded in Arabidopsis followed by confirming similar mechanisms are in place in monocot crop plants such as rice and Brachypodium.

Using a thorough methodology including genetic analysis, phenotyping, metabolic analysis, kinetic assays to measure enzyme activities and modelling crystal structures they provide an impressive amount of data to demonstrate that auxin degradation is a linear pathway comprising of initial step of conjugation to amino acids, followed by oxidation and then removal of the amino acid. Oxidation by DAOs is the first irreversible step.

Importantly this novel understanding of the auxin degradation pathway is of high significance to the field. This work ties up some loose ends as some published data did not make sense with the previous model. Given the importance of auxin as signalling molecules in plants, which based on cellular concentrations regulates almost every step of development, understanding how cells remove excess auxin will significantly contribute to understanding how cells can maintain and modify their auxin levels and thereby how one molecule can regulate such a myriad of developmental decisions.

However I do have some comments for the authors to address

>Response: Thank you so much for favorable comments on our manuscript. We have revised our manuscript according to the comments.

1. Kakeimide presents a novel inhibitor for GH3 enzymes and I am sure this will become an important and widely used inhibitor in the field of auxin homeostasis. While the molecular structure and synthesis instructions are provided, Authors do not comment about how this was designed, tested, etc. They state this will be published elsewhere but they should provide at least the author names so this will be searchable upon publication.

>Response: Thank you for your critical comments. We are now preparing a manuscript for the design and characterization of the GH3 inhibitor, kakeimide. According to the suggestions by editor, you, and reviewer #3, we attached the preliminary draft about kakeimide and will deposit the manuscript to a preprint server soon.

As you may know, the first inhibitor for GH3 acyl amino acid synthase was reported in 2012 (PLoS ONE 7(5):e37632). AIEP is an analog of IAA-AMP, an intermediate of GH3-catalyzed reaction. AIEP inhibits GH3 activity in vitro at 5-10 μ M range of K_i (KKI

A: X = CH₂, Adenosine-5'-[2-(1H-indol-3-yl)ethyl]phosphate (AIEP).
B: X = CO, Adenosine-5'-[2-(1H-indol-3-yl)acetyl]phosphate.

showed pico – nano M of K_i to GH3). We confirmed AIEP was found to be completely inactive in planta in our laboratory conditions. Another GH3 inhibitor targets the group I GH3 enzyme, GH3-11/JAR1 that function in jasmonate signaling (Nat. Chem Biol 2014 DOI: 10.1038/nchembio.1591). GH3-11/JAR1 catalyzes the conjugation of jasmonate and isoleucine to produce the active JA-Ile conjugate. We initially thought that it would be nearly impossible to find potent inhibitors that completely repress the IAA-conjugating activity of group II GH3 enzymes in vivo. Because GH3 is rapidly and highly up-regulated by its substrate IAA, suggesting that the inhibition of auxin-responsive GH3 enzymes would lead to the IAA accumulation and then the IAA-induced GH3 enzymes would overcome the inhibition by chemical inhibitors. Additionally, GH3 consists of a large gene family and using various substrates including JA, salicylic acid, IAA, and ABA, thus specificity to IAA-conjugating GH3 is highly important. Therefore, the design of in vivo effective inhibitor of group II GH3 would be highly challenging. In our draft, we determined the mode of inhibition, selectivity to GH3 enzymes, phenotypic analysis of kakeimide, GH3 inhibitor. We achieved temporal inhibition of IAA-inactivation driven by GH3. Very surprisingly, endogenous IAA level is rapidly elevated by GH3 inhibitor. The amount of endogenous IAA is almost doubled within 10 min, suggesting that turnover rate of endogenous IAA in steady state is unexpectedly fast. Thus, IAA-inactivation by GH3-DAO pathway would play a role in the regulation of rapid auxin responses. We hope that we have adequately addressed your concerns without providing additional experimental results because the paper is already data-intensive.

We reformatted the sections “Synthesis of kakeimide”, “Inhibitory activity of kakeimide on AtGH3.6” and table S3 “ K_i values of GH3 inhibitor, kakeimide on AtGH3.6”. in the supplemental methods and The preparation of recombinant GH3.6 protein was briefly documented in the methods section “Recombinant GH3.6 protein was purified by TALON metal affinity resin from the culture lysate of *E. coli* BL21 harboring pCold-GH3.6.”

2. Line 161: not IAA biosynthesis mutants have been used but wildtype was treated with IAA synthesis inhibitors Kyn + YDF Fig S7d

>Response: Reviwer #2 also commented on Fig S7d. We replaced Fig. S7d to new results (Fig. S7c and Fig. 3). We removed original Fig. S7d.

Original Fig. S7

Revised Fig. S7

In Fig. S7d, we would like to demonstrate that IAA-Asp-DM is metabolically converted to IAA and then restored IAA deficient phenotype in WT, *dao1*, *sav3-1* treated with auxin biosynthesis inhibitors (Original Fig. S7d). As you suggested by reviewers #1 and #2, the direct evidence for the metabolic conversion of IAA-Asp/IAA-Glu to IAA in vivo would be crucial for our work. We performed the feeding experiment in WT and *ilr1/ill* mutants using $^2\text{H}_2$ -labelled IAA-Asp-DM and $^2\text{H}_2$ -IAA-Glu-DM. $^2\text{H}_2$ -labelled IAA-Asp/Glu and $^2\text{H}_2$ -labelled IAA were measured by LC-MS/MS. This experiment directly indicates that the metabolic conversion of IAA-Asp/IAA-Glu into IAA by ILR1/ILL enzymes in vivo. These results are shown in revised Fig. 3c and original Fig. 3c was revised and moved to Fig. S7c.

3. It would be useful if the authors could provide some explanation or hypothesis on why *gh3oct/wt+KKI* (Fig S 3) can still produces DioxIAA

Fig.2c

Fig.S3

>Response: Thank you for your very crucial comments. Same issue was suggested by reviewer #1.

The oxIAA and dioxIAA level in *gh3-oct* (*gh3-sept*) mutants were lower than those in WT, but the oxIAA and dioxIAA level are still considerably higher than the level in *ilr1* and *dao1dao2* mutants. Our preliminary experiment found that KKI reduced only 25% of oxIAA in WT plants grown with 10 μM KKI for 8 days. In contrast, in short-term treatment (20 h), KKI potentially inhibited oxIAA production from exogenous IAA application or in IAA overproduction *YUC2-ox* line (Fig. 2d and Fig.S3d). As you know,

IAA is unstable and non-enzymatically oxidized to oxIAA. However, oxIAA level in *ilr1* mutants remain low level even after IAA treatment (Fig. 6d). These results suggest that most of oxIAA is produced by DAO1 and ILR1. We performed additional feeding experiments with ^{13}C -labeled IAA. 7-days-old WT, *dao1 dao2*, *gh3-oct* (*gh3-sept*) seedlings were incubated with ^{13}C -labelled IAA for 20 h. The ^{13}C -oxIAA and ^{13}C -dioxIAA were quantified by LC-MS/MS. The results clearly demonstrate that exogenous $^{13}\text{C}_6$ -IAA was not converted to $^{13}\text{C}_6$ -oxIAA and $^{13}\text{C}_6$ -dioxIAA in *dao1 dao2* and *gh3-sept* mutants, but did convert in WT plants, indicating IAA is oxidized to oxIAA via IAA-amino acid conjugates. We also examined the effects of KKI on oxIAA level in *dao1 dao2* mutants. The *dao1dao2* mutants were grown with KKI for 8 days that phenocopied high auxin phenotypes in *gh3-sept* mutants. The oxIAA level in *dao1dao2* mutants remains unchanged after KKI treatment for 8days. Thus, it is unlikely that oxIAA is directly produced from IAA by non-enzymatic oxidation of IAA. We speculate that the group III GH3 enzymes, such as AtGH3.15 having low affinity to IAA might produce IAA-amino acid conjugates at high IAA condition in *gh3-sept* and thereby storing small amounts of oxIAA. Indeed, Indeed, AtGH3.15 showed the K_{cat}/K_m value of $23 \text{ m}^{-1} \text{ s}^{-1}$ for IAA, and this K_{cat}/K_m value was 7% of the value of AtGH3.5 ($314 \text{ m}^{-1} \text{ s}^{-1}$) (J.Biol.Chem, 293, 2018, 4277-4288, Table 2 and 3). These issue are discussed in revised draft (line 287-301).

Revised Fig 2e

(Fig. 2e) $^{13}\text{C}_6$ IAA feeding experiment. $^{13}\text{C}_6$ oxIAA and $^{13}\text{C}_6$ dioxIAA were measured after incubation of 7-d-old WT, *dao1dao2*, and *gh3-sept* seedlings with $0.2 \mu\text{M}$ $^{13}\text{C}_6$ IAA for 20 h.

Fig R4.

Fig. 6d

In Fig. R4 (for the review), WT and *dao1dao2* mutants were grown for 8 days with KKI (5 μ M). The WT and *dao1 dao2* mutants showed high auxin phenotype like *gh3-sept* mutants. LC-MS/MS analysis indicated that oxIAA level in *dao1dao2* remain unchanged at low level. This result suggests that oxIAA was produced from non-enzymatic oxidation from IAA. Fig.6d indicated that oxIAA level was not increased after IAA treatment in *ilr1* mutants, suggesting most of oxIAA is produced IAA-amino acid conjugates.

(Revised, line 289)

The metabolic analysis of *ilr1* mutants and kinetic study for ILR1 enzyme revealed that ILR1 converts oxIAA-Asp and oxIAA-Glu to oxIAA in *Arabidopsis* plants (Fig. 6 and Supplementary Fig.19 and 20). Our feeding experiments showed that oxIAA and dioxIAA level in *ilr1* mutants remain unchanged even after IAA treatment (Fig. 6d). Similarly, previous work demonstrated that oxIAA was not produced from exogenous IAA in *dao1-1* mutants¹⁰. These metabolite analyses firmly established that ILR1 is essential for oxIAA production and DAO1 did not directly convert IAA to oxIAA. This evidence suggests that most of oxIAA is produced from IAA-amino acid conjugates by DAO1 and ILR1 enzymes, but not by enzymatic or non-enzymatic direct oxidation of IAA. Although the feeding experiment showed that ¹³C-oxIAA is not synthesized from ¹³C-labeled IAA in *gh3-sept* mutants during 20 h incubation (Fig. 2e), *gh3-sept* mutant still accumulates endogenous oxIAA (30-40 % amounts in WT plants) that were higher than the levels in the *dao1 dao2* and *ilr1* mutants (Fig. 2c). At a high level of IAA in *gh3-sept* mutants, the group III GH3 enzymes, such as AtGH3.15 might produce IAA-amino acid conjugates to compensate an impaired IAA inactivation in *gh3-sept* mutants, thereby accumulating small amounts of oxIAA.

4. Fig S5: Why has this not been crystallised/modelled with IAA-Glu, given the importance of this conjugate

>Response: Thank you for your crucial suggestions. According to your suggestion, we replaced the molecular docking of IAA-Leu to that of IAA-Glu in Fig. S5.

5. Fig S9a: It's not clear with what the imaged seedlings have been treated

>Response: Thank you for your suggestions. Same question was raised by reviewer #2. This was our error. We revised Figure S9a as indicated below.

REVIEWERS' COMMENTS

Reviewer #1 (Remarks to the Author):

The authors have addressed my previous concerns very well, and have added important new results.

My only remaining question concerns how [¹³C₆]oxIAA was measured to give the results shown in Fig 2e. It appears that the internal standard used to measure oxIAA is itself [¹³C₆] -labelled, and the question therefore arises, how could the authors distinguish between the [¹³C₆]oxIAA produced (indirectly) from [¹³C₆]IAA and the [¹³C₆] -labelled internal standard? The dideuterated internal standard used for dioxIAA would not pose a problem, because the mass spectrometer could distinguish the deuterated and [¹³C₆] forms. I am just asking for a sentence in the methods section to clarify this issue.

I also recommend the following minor changes:

Line 76: Change ".....IAA-Glu in plants have not been experimentally....." to ".....IAA-Glu in plants has not been experimentally".

Line 168: Change ".....IAA-Asp-DM and IAA-Glu-dimethyl ester (IAA-Glu-DM) would be metabolically converted to" to ".....IAA-Asp-DM and IAA-Glu-dimethyl ester (IAA-Glu-DM) were metabolically converted to".

Lines 247-248: Change "This evidence indicates that high-auxin phenotype of dao1-1 mutants would attribute to IAA release from IAA-Asp and IAA-Glu by ILR1/ILLs." to "This evidence indicates that the high-auxin phenotype of dao1-1 mutants can be attributed to IAA release from IAA-Asp and IAA-Glu by ILR1/ILLs."

Line 295: Change the wording to: "...for oxIAA production and that DAO1 does not directly convert".

Line 548: Change "are analysed" to "were analysed".

John Ross

Reviewer #2 (Remarks to the Author):

A majority of my original comments have been addressed. I am satisfied with the revisions made. A few minor English language edits are necessary, as indicated below.

Line 76, change "have not" to "has not", "did not show" to "did not trigger".

Line 82, "of the conjugates".

Line 122-123, change "would be highly accumulated" to "would accumulate to high levels".

Line 126, add a reference after oxIAA.

Line 130, change "but abundantly converted" to "but was efficiently converted".

Line 168, add a comma after "(IAA-Glu-DM)"; change "would be" to "are".

The term pro-drug is sometimes used with and sometimes without a dash. Reconcile the spelling.

Line 177, "are the substrates".

Line 239, change "than dao1-1" to "than did dao1-1".

Line 246-247, change "high-auxin phenotype of dao1-1 mutants would attribute to IAA release" to "high-auxin phenotypes of dao1-1 mutants are the consequence of the IAA release".

Line 257, it should be "metabolite analysis" (singular).

Lines 258-259, change "endogenous oxIAA and dioxIAA were dramatically decreased, and oxIAA-Glu greatly accumulated in the two ilr1 mutants" to "the levels of endogenous oxIAA and dioxIAA dramatically decreased, and whereas oxIAA-Glu greatly accumulated in the two ilr1 mutants".

Line 548, "were analyzed".

Reviewer #3 (Remarks to the Author):

The revised manuscript by Hayashi et al is an impressive and data heavy manuscript on auxin degradation. The authors provide exhaustive but compelling evidence that auxin conjugation to amino acids followed by oxidation is the major linear pathway for removing and thereby regulating cellular auxin concentrations. The authors did a great job revising the manuscript in response to my and the other Reviewer's comments.

While the manuscript is still difficult to read due to the amount of data presented I believe that this level of evidence is needed, and that this is now well presented in a logic way.

I leave it to the Editor to decide if and how this can be published while the KKI manuscript is not available, but for me the summary of the 3 presented manuscript provided sufficient and compelling evidence and I am looking forward to seeing this manuscript published.

Some minor comments:

Line 72: change conjugate to conjugates

Line 97: this should be KKI...showed competitive inhibition of IAA-conjugation

Line 220: statement refers to Suppl Fig 15 (instead of 14?!))

Line 351: change form to from

Reviewer #4 (Remarks to the Author):

Review of the synthesis of kakeimide

The authors described the synthesis of kakeimide in two steps from phthalic anhydride. The first reaction was performed following an experimental protocol described in the reference Guenin, 2007 #50 (Eur. J. Org. 2007, 3380-3391), however this reference is missing both in the main manuscript and in the supporting information document.

High resolution mass spectrometry data for 4-(1,3-dioxoisindolin-2-yl)butanoic acid is missing. Being a solid, the melting point should be provided and compared to that of the literature (Eur. J. Org. 2007, 3380-3391).

In the ¹H NMR description of this compound the signals 7.85 (q, J = 2.7 Hz, 2H), 7.72 (q, J = 2.7 Hz, 2H), corresponding to the aromatic protons can not be described as quartets, as any of these protons have 3 chemically equivalent neighbouring protons. Thus the quartet attribution should be replaced by multiplet.

Figures of the ¹H and ¹³C NMR spectra are missing for this compound.

Regarding the second reaction, the preparation of kakeimide from 4-(1,3-dioxoisindolin-2-yl)butanoic acid, the full name of the reagent WSDC-HCl should be provided when it is first

mentioned.

During the extraction with ethyl acetate, how the pH was brought to/was maintained at 2-3?

In the ^1H NMR description of kakeimide, the signals 7.84 (q, $J = 2.9$ Hz, 2H), 7.71 (q, $J = 2.9$ Hz, 2H), corresponding to aromatic protons can not be described as quartets, as any of these protons have 3 chemically equivalent neighbouring protons.

The signal at 2.12 is not a quartet but a multiplet.

In ^{13}C NMR, the signals at 77.3, 77.0, 76.7 does not belong to kakeimide but to the CHCl_3 present in CDCl_3 used as the deuterated solvent, thus these signals need to be removed from the ^{13}C NMR description.

High resolution mass spectrometry data for kakeimide is missing.

The figures of the ^1H and ^{13}C NMR spectra could be larger.

Other comments:

Fig.1 a, compounds oxIAA, DioxIAA and oxIAA-Glu have new asymmetric carbons, which is the configuration of these carbons? The authors should represent it in the structure of the compounds.

Reviewer #1

The authors have addressed my previous concerns very well, and have added important new results.

My only remaining question concerns how [$^{13}\text{C}_6$]oxIAA was measured to give the results shown in Fig 2e. It appears that the internal standard used to measure oxIAA is itself [$^{13}\text{C}_6$] -labelled, and the question therefore arises, how could the authors distinguish between the [$^{13}\text{C}_6$]oxIAA produced (indirectly) from [$^{13}\text{C}_6$]IAA and the [$^{13}\text{C}_6$] -labelled internal standard? The dideuterated internal standard used for dioxIAA would not pose a problem, because the mass spectrometer could distinguish the deuterated and [$^{13}\text{C}_6$] forms. I am just asking for a sentence in the methods section to clarify this issue.

>Thank you for your valuable suggestions. We are thankful for the time and energy you expended. As you mentioned, we used [$^2\text{H}_2$]oxIAA and [$^2\text{H}_2$]DioxIAA as internal standards for the experiments in Fig.2e.

We revised as follow in Fig. 2 legend:

(e) [$^{13}\text{C}_6$]IAA feeding experiment. [^{12}C]-endogenous and [$^{13}\text{C}_6$]-labeled oxIAA and dioxIAA were measured using [$^2\text{H}_2$]oxIAA and [$^2\text{H}_2$]dioxIAA as internal standards after incubation of 7-d-old WT, *dao1dao2*, and *gh3-sept* seedlings with 0.2 μM [$^{13}\text{C}_6$]IAA for 20 h.

I also recommend the following minor changes:

>I will revise the text according to the suggestions

Line 76: Change “.....IAA-Glu in plants have not been experimentally.....” to “.....IAA-Glu in plants has not been experimentally”.

>revised as suggested.

Line 168: Change “.....IAA-Asp-DM and IAA-Glu-dimethyl ester (IAA-Glu-DM) would be metabolically converted to” to “.....IAA-Asp-DM and IAA-Glu-dimethyl ester (IAA-Glu-DM) were metabolically converted to”.

> According to the suggestion by reviewers #1 and #2, we revised to “IAA-Asp-DM and IAA-Glu-dimethyl ester (IAA-Glu-DM), were metabolically converted, to the unmodified IAA-Asp and IAA-Glu,”

Lines 247-248: Change “This evidence indicates that high-auxin phenotype of *dao1-1* mutants would attribute to IAA release from IAA-Asp and IAA-Glu by ILR1/ILLs.” to “This evidence indicates that the high-auxin phenotype of *dao1-1* mutants can be attributed to IAA release from IAA-Asp and IAA-Glu by ILR1/ILLs.”

> According to the suggestion by reviewers #1 and #2, we revised to “This evidence indicates that the high-auxin phenotypes of *dao1-1* mutants are the consequence of the IAA release from IAA-Asp and IAA-Glu by ILR1/ILLs.”

Line 295: Change the wording to: “...for oxIAA production and that DAO1 does not directly convert”.

> Revised as suggested. “ILR1 is essential for oxIAA production and that DAO1 did not directly convert IAA to oxIAA.”

Line 548: Change “are analysed” to “were analysed”.

>revised to “ were analyzed”.

Reviewer #2 (Remarks to the Author):

A majority of my original comments have been addressed. I am satisfied with the revisions made. A few minor English language edits are necessary, as indicated below.

>Response: Thank you very much for providing comments. We are thankful for the time and energy you expended. We have revised our manuscript accordingly.

Line 76, change “have not” to “has not”, “did not show” to “did not trigger”.

> Revised as suggested. “both conjugates did not trigger auxin-phenotypes”

Line 82, “of the conjugates”.

> Revised as suggested

Line 122-123, change “would be highly accumulated” to “would accumulate to high levels”.

> Revised as suggested. “we expected that oxIAA would accumulate to high levels in *gh3-sept* mutants.”

Line 126, add a reference after oxIAA.

>Reference 22. “Reinecke DM, Bandurski RS. Oxidation of indole-3-acetic acid to oxindole-3-acetic acid by an enzyme preparation from *Zea mays*. *Plant Physiol* 86, 868-872 (1988).” was cited.

Line 130, change “but abundantly converted” to “but was efficiently converted”.

> Revised as suggested

Line 168, add a comma after "(IAA-Glu-DM)"; change "would be" to "are".

> According to the suggestion by reviewers #1 and #2, we revised to "IAA-Asp-DM and IAA-Glu-dimethyl ester (IAA-Glu-DM), were metabolically converted to the unmodified IAA-Asp and IAA-Glu,"

The term pro-drug is sometimes used with and sometimes without a dash. Reconcile the spelling.

> revised "pro-drug" to "produg".

Line 177, "are the substrates".

> revised as suggested.

Line 239, change "than dao1-1" to "than did dao1-1".

> revised as suggested

Line 246-247, change "high-auxin phenotype of dao1-1 mutants would attribute to IAA release" to "high-auxin phenotypes of dao1-1 mutants are the consequence of the IAA release".

> According to the suggestion by reviewers #1 and #2, we revised to "This evidence indicates that high-auxin phenotype of *dao1-1* mutants are the consequence of the IAA release from IAA-Asp and IAA-Glu by ILR1/ILLs."

Line 257, it should be "metabolite analysis" (singular).

> Revised as suggested. "our metabolite analysis demonstrated that"

Lines 258-259, change "endogenous oxIAA and dioxIAA were dramatically decreased, and oxIAA-Glu greatly accumulated in the two *ilr1* mutants" to "the levels of endogenous oxIAA and dioxIAA dramatically decreased, and whereas oxIAA-Glu greatly accumulated in the two *ilr1* mutants".

> Revised as suggested. "our metabolite analysis demonstrated that the levels of endogenous oxIAA and dioxIAA dramatically decreased, and whereas oxIAA-Glu greatly accumulated in the two *ilr1* mutants"

Line 548, "were analyzed".

> Revised as suggested.

Reviewer #3 (Remarks to the Author):

The revised manuscript by Hayashi et al is an impressive and data heavy manuscript on auxin degradation. The authors provide exhaustive but compelling evidence that auxin conjugation to amino acids followed by oxidation is the major linear pathway for removing and thereby regulating cellular auxin concentrations. The authors did a great job revising the manuscript in response to my and the other Reviewer's comments.

While the manuscript is still difficult to read due to the amount of data presented I believe that this level of evidence is needed, and that this is now well presented in a logic way.

I leave it to the Editor to decide if and how this can be published while the KKI manuscript is not available, but for me the summary of the 3 presented manuscript provided sufficient and compelling evidence and I am looking forward to seeing this manuscript published.

>Response: Thank you very much for providing comments. We are thankful for the time and energy you expended. We have revised our manuscript accordingly.

Some minor comments:

Line 72: change conjugate to conjugates.

Line 72, genetic screen for mutants resistant to IAA-Leu conjugate.

>We revised "IAA-Leu conjugate" to "IAA-Leu".

Line 97: this should be KKI...showed competitive inhibition of IAA-conjugation

>KKI competitively bind IAA binding site of GH3 and then inhibit the formation of GH3 reaction intermediate, IAA-AMP from IAA and ATP. We will deposit two manuscripts on kakeimide and gh3 mutant on preprint server.

Line 220: statement refers to Suppl Fig 15 (instead of 14?!)

>We revised to "Among the single mutants, only *ilr1* showed resistance to IAA-Asp-DM and IAA-Glu-DM in primary root growth and lateral root promotion (Fig. 5a and Supplementary Fig. 14 and 15d)."

Line 351: change form to from

>revised as suggested.

Reviewer #4

Review of the synthesis of kakeimide

>Thank you for your valuable comments on the synthesis of kakeimide. We are thankful for the time and energy you expended.

The authors described the synthesis of kakeimide in two steps from phthalic anhydride. The first reaction was performed following an experimental protocol described in the reference Guenin, 2007 #50 (Eur. J. Org. 2007, 3380-3391), however this reference is missing both in the main manuscript and in the supporting information document.

>Please see "References and Notes section", Ref No.5 (Eur. J. Org. 2007, 3380-3391) at the last page (Page 38) in Supplementary Information file. The table of contents was shown in the first page of Supplementary Information file.

References and Notes

1. Curtis, M.D. & Grossniklaus, U. A gateway cloning vector set for high-throughput functional analysis of genes in planta. *Plant Physiol* **133**, 462-469 (2003).
2. Nakagawa, T. *et al.* Development of series of gateway binary vectors, pGWBs, for realizing efficient construction of fusion genes for plant transformation. *J Biosci Bioeng* **104**, 34-41 (2007).
3. Narusaka, M., Shiraishi, T., Iwabuchi, M. & Narusaka, Y. The floral inoculating protocol : a simplified *Arabidopsis thaliana* transformation method modified from floral dipping. *Plant biotechnology* **27**, 349-351 (2010).
4. Trott, O. & Olson, A.J. AutoDock Vina: improving the speed and accuracy of docking with a new scoring function, efficient optimization, and multithreading. *J Comput Chem* **31**, 455-461 (2010).
5. Guenin, E., Monteil, M., Bouchemal, N., Prange, T. & Lecouvey, M. Syntheses of phosphonic esters of alendronate, pamidronate and neridronate. *European Journal of Organic Chemistry* **2007**, 3380-3391 (2007).
6. Chen, Q., Westfall, C.S., Hicks, L.M., Wang, S. & Jez, J.M. Kinetic basis for the conjugation of auxin by a GH3 family indole-acetic acid-amido synthetase. *J Biol Chem* **285**, 29780-29786 (2010).

High resolution mass spectrometry data for 4-(1,3-dioxoisindolin-2-yl)butanoic acid is missing. Being a solid, the melting point should be provided and compared to that of the literature (Eur. J. Org. 2007, 3380-3391). Figures of the 1H and 13C NMR spectra are missing for this compound.

>Thank you for your comments. The compound, 4-(1,3-dioxoisindolin-2-yl)butanoic acid [CAS 3130-75-4] is known compounds and commercially available as reagents from Merck-Aldrich and Tokyo Chemical Industry, etc (90 chemical suppliers of this reagent, and 53 reactions were reported previously). We synthesized this compound because this chemical is relatively expensive. Thus, we omitted the high-MS data, melting point, and NMR spectra according to the general rule in organic chemistry article.

Reactions (94)

View in SciFinder[®]

Scheme 1 (53 Reactions)

Steps: 1 Yield: 43-100%

Suppliers (70) Suppliers (125) Suppliers (90)

Reaction Summary	Steps: 1 Yield: 100%	4-Substituted 2-amino-3,4-dihydroquinazolines with a 3-hairpin turn side chain as novel inhibitors of BACE-1 By: Jagtap, Ajit Dhananjay; et al Bioorganic Chemistry (2020), 95, 103135.
1.1 6 h, 150 °C		
Experimental Protocols		
Reaction Summary	Steps: 1 Yield: 100%	Syntheses of phosphonic esters of alendronate, pamidronate and neridronate By: Guenin, Erwann; et al European Journal of Organic Chemistry (2007), (20), 3380-3391, S3380/1-S3380/55.
1.1 6 h, 170 °C		
Experimental Protocols		

In the 1H NMR description of this compound the signals 7.85 (q, J = 2.7 Hz, 2H), 7.72 (q, J = 2.7 Hz, 2H), corresponding to the aromatic protons can not be described as quartets, as any of these protons have 3 chemically equivalent neighbouring protons. Thus the quartet attribution should be replaced by multiplet.

>Thank you for your suggestions. These values are errors due to the automated signal picking. I revised as follow.

>original

¹H-NMR (400 MHz, CDCl₃) δ 7.85 (q, J = 2.7 Hz, 2H), 7.72 (q, J = 2.7 Hz, 2H), 3.77 (t, J = 6.9 Hz, 2H), 2.42 (t, J = 7.6 Hz, 2H), 2.05-1.98 (m, 2H);

>revised

¹H-NMR (400 MHz, CDCl₃) δ 7.85 (m, 2H), 7.72 (m, 2H), 3.77 (t, J = 6.9 Hz, 2H), 2.42 (t, J = 7.6 Hz, 2H), 2.05-1.98 (m, 2H);

Regarding the second reaction, the preparation of kakeimide from 4-(1,3-dioxoisindolin-2-yl) butanoic acid, the full name of the reagent WSCD-HCl should be provided when it is first mentioned.

>WSCD-HCl was changed to EDC-HCl as a more commonly used name.

WSCD-HCl > EDC-HCl, 1-ethyl-3-(3-dimethylaminopropyl)carbodiimide monohydrochloride

During the extraction with ethyl acetate, how the pH was brought to/was maintained at 2-3?

>original

The reaction mixture was poured into water (40 mL) and extracted with EtOAc (30 mL × 2 times) at pH 2-3.

>revised

The reaction mixture was poured into water (40 mL) and adjusted to pH 2-3 with 6M HCl. The mixture was extracted with EtOAc (30 mL × 2 times)

During the extraction with ethyl acetate, how the pH was brought to/was maintained at 2-3?

In the ¹H NMR description of kakeimide, the signals 7.84 (q, J = 2.9 Hz, 2H), 7.71 (q, J = 2.9 Hz, 2H), corresponding to aromatic protons can not be described as quartets, as any of these protons have 3 chemically equivalent neighbouring protons. The signal at 2.12 is not a quartet but a multiplet.

In ¹³C NMR, the signals at 77.3, 77.0, 76.7 does not belong to kakeimide but to the CHCl₃ present in CDCl₃ used as the deuterated solvent, thus these signals need to be removed from the ¹³C NMR description.

Thank you for your comments. These values are errors due to the automated signal picking.

>original

¹H-NMR (400 MHz, CDCl₃) δ 8.13 (s, 1H), 7.84 (q, J = 2.9 Hz, 2H), 7.71 (q, J = 2.9 Hz, 2H), 7.33 (s, 1H), 7.17 (t, J = 8.2 Hz, 1H), 7.02 (d, J = 8.7 Hz, 1H), 6.62 (d, J = 8.2 Hz, 1H), 4.56-4.50 (m, 1H), 3.81 (t, J = 6.2 Hz, 2H), 2.37 (t, J = 6.9 Hz, 2H), 2.12 (q, J = 6.6 Hz, 2H), 1.32 (d, J = 6.0 Hz, 6H); ¹³C-NMR (100 MHz, CDCl₃) δ 170.3, 168.9, 158.4, 139.2, 134.1, 131.9, 129.5, 123.3, 111.8, 111.6, 107.3, 77.3, 77.0, 76.7, 69.8, 37.1, 35.0, 25.2, 22.0;

>revised

¹H-NMR (400 MHz, CDCl₃) δ 8.13 (s, 1H), 7.84 (m, 2H), 7.71 (m, 2H), 7.33 (s, 1H), 7.17 (t, J = 8.2 Hz, 1H), 7.02 (d, J = 8.7 Hz, 1H), 6.62 (d, J = 8.2 Hz, 1H), 4.56-4.50 (m, 1H), 3.81 (t, J = 6.2 Hz, 2H), 2.37 (t, J = 6.9 Hz, 2H), 2.12 (m, 2H), 1.32 (d, J = 6.0 Hz, 6H); ¹³C-NMR (100 MHz, CDCl₃) δ 170.3, 168.9, 158.4, 139.2, 134.1, 131.9, 129.5, 123.3, 111.8, 111.6, 107.3, 69.8, 37.1, 35.0, 25.2, 22.0;

High resolution mass spectrometry data for kakeimide is missing.

>We measured high-resolution MS spectra again using the reflection mode of MS (high MS mode) and revised as follow in SI-method section.

>MALDI-TOFMS [M+Na]⁺ m/z calcd. for C₂₁H₂₂N₂O₄Na: 389.14773, found 389.14682.

Comment 1
Comment 2

Acquisition Parameter

Date of acquisition 2021-08-31T10:35:23.588+09:00
Acquisition method name D:\Methods\flexControlMethods\IRP_0_2kDa.par
Acquisition operation mode Reflector
Voltage polarity POS
Number of shots 1500
Name of spectrum used for calibration
Calibration reference list used POS_PEG-H-OH_Na_mono_KDO

Instrument Info

User BDAL@DE
Instrument FLEX-PC
Instrument type autoflexTOF/TOF

The figures of the ¹H and ¹³C NMR spectra could be larger.
>The NMR spectra will be enlarged in the documents.

Revised supplemental information (page 25). The revised parts are marked as red colored.

2.42 (t, $J = 7.6$ Hz, 2H), 2.05-1.98 (m, 2H); ^{13}C -NMR (100 MHz, CDCl_3) δ 178.5, 168.4, 134.0, 132.0, 123.3, 37.1, 31.3, 23.6.

To the solution of 4-(1,3-dioxoisindolin-2-yl) butanoic acid (940 mg, 4.0 mmol) in 8 mL of DMF was added 1-hydroxybenzotriazole monohydrate (545 mg, 4.0 mmol), 3-isopropoxyaniline (610 mg, 4.0 mmol), and EDC-HCl, **1-Ethyl-3-(3-dimethylaminopropyl)carbodiimide monohydrochloride** (773 mg, 4.0 mmol). The reaction mixture was stirred for 2 h at room temperature. The reaction mixture was poured into water (40 mL) and adjusted to pH 2-3 with 6M HCl. The mixture was extracted with EtOAc (30 mL \times 2 times) and the EtOAc layer was then washed with 1 M Na_2CO_3 aqueous solution (30 mL). The EtOAc layer was dried over anhydrous Na_2SO_4 and then concentrated *in vacuo*. The resulting powder was recrystallized from *n*-hexane-EtOAc. The product, kakeimide, (4-(1,3-dioxoisindolin-2-yl)-*N*-(3-isopropoxyphenyl) butanamide) was obtained as a colorless powder (1007 mg, 68% yield). m.p.= 113–115°C; ^1H -NMR (400 MHz, CDCl_3) δ 8.13 (s, 1H), **7.84 (m, 2H)**, **7.71 (m, 2H)**, 7.33 (s, 1H), 7.17 (t, $J = 8.2$ Hz, 1H), 7.02 (d, $J = 8.7$ Hz, 1H), 6.62 (d, $J = 8.2$ Hz, 1H), 4.56-4.50 (m, 1H), 3.81 (t, $J = 6.2$ Hz, 2H), 2.37 (t, $J = 6.9$ Hz, 2H), 2.12 (m, 2H), 1.32 (d, $J = 6.0$ Hz, 6H); ^{13}C -NMR (100 MHz, CDCl_3) δ 170.3, 168.9, 158.4, 139.2, 134.1, 131.9, 129.5, 123.3, 111.8, 111.6, 107.3, 69.8, 37.1, 35.0, 25.2, 22.0; **MALDI-TOFMS $[\text{M}+\text{Na}]^+$ m/z calcd. for $\text{C}_{21}\text{H}_{22}\text{N}_2\text{O}_4\text{Na}$: 389.14773, found 389.14682.**

Other comments:

Fig.1 a, compounds oxIAA, DioxIAA and oxIAA-Glu have new asymmetric carbons, which is the configuration of these carbons? The authors should represent it in the structure of the compounds.

Supplemental Fig. 4 (g) oxIAA-amino acids were detected as a diastereomeric mixture from the reaction mixture. One of the diastereomers of oxIAA-Asp was readily epimerized during sample preparation.

As shown in Supplemental Fig.4 g, oxIAA-Asp are readily epimerized to yield diastereomeric mixture (*R*-oxIAA-L-Asp and *S*-oxIAA-L-Asp). oxIAA was also rapidly epimerized. The AtDAO1 enzyme stereoselectively oxidize IAA-L-amino acids to yield enantiomer of oxIAA-L-amino acids. However, the enantiomer is epimerized soon to be the mixture of diastereomers. Thus, naturally occurring oxIAA and DioxIAA would be present as the racemic

forms in planta or would be difficult to determine the predominant enantiomeric form of these compounds. Thus, we indicated the structures of oxIAA, DioxIAA and oxIAA-Glu as racemic forms in Fig. 1a.